# PGN: The RNN's New Successor is Effective for Long-Range Time Series Forecasting

**Yuxin Jia**[1,2]   **Youfang Lin**[1,2]   **Jing Yu**[1,2]   **Shuo Wang**[1,2]   **Tianhao Liu**[3]   **Huaiyu Wan**[1,2,*]

[1] School of Computer Science & Technology, Beijing Jiaotong University, China
[2] Beijing Key Laboratory of Traffic Data Analysis and Mining, Beijing, China
[3] School of Cyberspace Science and Technology, Beijing Jiaotong University, China
{yuxinjia, yflin, jingyu1, shuo.wang, leolth, hywan}@bjtu.edu.cn

## Abstract

Due to the recurrent structure of RNN, the long information propagation path poses limitations in capturing long-term dependencies, gradient explosion/vanishing issues, and inefficient sequential execution. Based on this, we propose a novel paradigm called Parallel Gated Network (PGN) as the new successor to RNN. PGN directly captures information from previous time steps through the designed Historical Information Extraction (HIE) layer and leverages gated mechanisms to select and fuse it with the current time step information. This reduces the information propagation path to $\mathcal{O}(1)$, effectively addressing the limitations of RNN. To enhance PGN's performance in long-range time series forecasting tasks, we propose a novel temporal modeling framework called Temporal PGN (TPGN). TPGN incorporates two branches to comprehensively capture the semantic information of time series. One branch utilizes PGN to capture long-term periodic patterns while preserving their local characteristics. The other branch employs patches to capture short-term information and aggregate the global representation of the series. TPGN achieves a theoretical complexity of $\mathcal{O}(\sqrt{L})$, ensuring efficiency in its operations. Experimental results on five benchmark datasets demonstrate the state-of-the-art (SOTA) performance and high efficiency of TPGN, further confirming the effectiveness of PGN as the new successor to RNN in long-range time series forecasting. The code is available in this repository: https://github.com/Water2sea/TPGN.

## 1 Introduction

Under the premise of accurate time series forecasting, long-range forecasting tasks offer an advantage over short-range forecasting tasks as they provide more comprehensive information for individuals and organizations to thoroughly assess future changes and make well-informed decisions. Due to its practical applicability across various fields (i.e., energy [Zhou et al., 2021], climate [Angryk et al., 2020], traffic [Yin and Shang, 2016], etc), long-range forecasting has attracted significant attention from researchers in recent years.

Long-range time series forecasting tasks can be broadly classified into two categories. One task is to utilize abundant inputs to forecast future outputs [Liu et al., 2022a, Jia et al., 2023], while another task is to predict longer-range futures with fewer historical inputs [Zhou et al., 2021, 2022a, Wang et al., 2023]. Although existing studies have shown that ample historical inputs can introduce more information to improve prediction performance [Jia et al., 2023, Liu et al., 2024], considering factors such as the load capacity of training devices and data collection, the utilization of limited historical

---

*Corresponding author

38th Conference on Neural Information Processing Systems (NeurIPS 2024).

inputs to predict longer-range futures remains an important research topic. Therefore, this paper sets the task goal as predicting longer outputs with fewer inputs.

In recent years, deep-learning-based methods have achieved remarkable success in time series forecasting (for further discussions, please refer to Section 2 and Appendix B). These methods can be roughly categorized into four based paradigms: **Transformers** [Zhou et al., 2021, Wu et al., 2021, Liu et al., 2022a, Zhou et al., 2022a, Nie et al., 2023, Ni et al., 2023, Liu et al., 2024, Dai et al., 2024], **CNNs** [Wu et al., 2023, Wang et al., 2023, Luo and Wang, 2024], **MLPs and Linears** [Zeng et al., 2023, Xu et al., 2024, Wang et al., 2024], and **RNNs** [Jia et al., 2023]. It is worth noting that RNNs have received relatively less attention over an extended period of time. **This discrepancy is primarily attributed to the limitation of RNNs' recurrent structure, which leads to the persistence of excessive long pathways for information propagation.**

In fact, shorter information propagation paths lead to less information loss [Tishby and Zaslavsky, 2015], better captured dependencies [Liu et al., 2022a], and lower training difficulty [Wang et al., 2023]. However, RNNs heavily rely on a sequential recurrent structure to transmit information, making it challenging for them to capture long-term dependencies and suffer from the issue of gradient vanishing/exploding [Pascanu et al., 2013]. Meanwhile, due to its sequential computation, even though RNNs have a theoretical complexity that is linear with respect to sequence length $L$, their actual running speed can be even slower than the $\mathcal{O}(L^2)$ complexity of the Vanilla-Transformer [Vaswani et al., 2017]. Some RNN-based models [Hochreiter and Schmidhuber, 1997, Chung et al., 2014] have tried to enhance performance by incorporating specialized gated mechanisms. However, compared to the inherent limitations of the RNN structure, these improvements in information selection and fusion are merely a drop in the bucket.

Based on this motivation, this paper proposes a novel general paradigm called **P**arallel **G**ated **N**etwork (**PGN**) **as the new successor to RNN**. PGN introduces a Historical Information Extraction (HIE) layer to replace the recurrent structure of RNN, and then further selects and fuses information through gated mechanisms, effectively reducing the information propagation paths to $\mathcal{O}(1)$, as shown in Figure 1 (l). This enables PGN to better capture long-term dependencies in input signals. Additionally, since computations for each time step can be parallelized, PGN achieves significantly faster execution speed while maintaining the same theoretical complexity of $\mathcal{O}(L)$ as RNN.

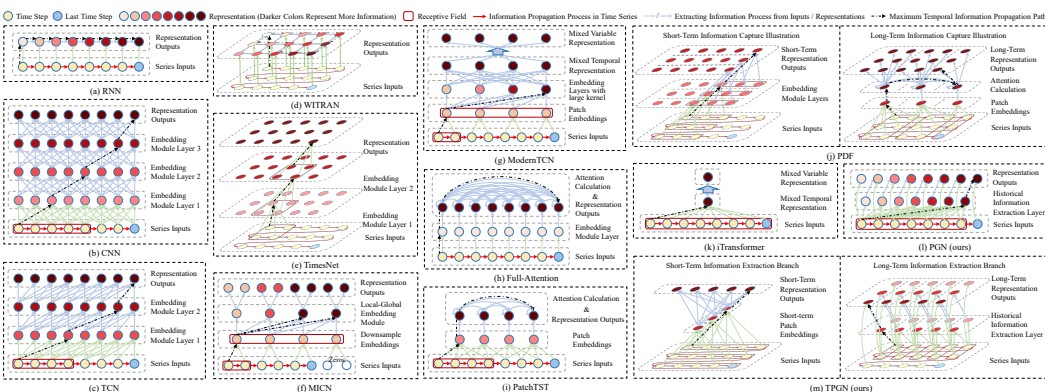

Figure 1: The information propagation illustration of different models.

Despite the advantages of PGN in terms of efficiency and capturing long-term information, it cannot be directly applied to time series forecasting tasks for optimal performance. This is because, based on 1D modeling, PGN struggles to capture periodic semantic information [Jia et al., 2023] effectively. Fortunately, the idea of transforming data from 1D to 2D and modeling it [Wu et al., 2023, Jia et al., 2023, Dai et al., 2024], proves effective in addressing above limitation. When employing 2D modeling for time series, information captured along rows reflects short-term changes, while information along columns represents long-term periodic patterns. Due to the distinct characteristics of these two types of information, it is reasonable to model them separately. Furthermore, considering that periodicity is present throughout the entire time series, both in the past and in the future, it is important to prioritize this consideration when modeling.

Based on these motivations, we propose a novel PGN-based temporal modeling framework called **T**emporal **P**arallel **G**ated **N**etwork (**TPGN**). TPGN establishes two distinct branches to capture long-term and short-term information in the 2D input series. To focus on modeling long-term information, we utilize PGN to model each column of the 2D inputs, preserving their respective local periodic characteristics. Simultaneously, leveraging the advantages of patch [Nie et al., 2023] in capturing short-term changes, TPGN initially aggregates the short-term information into patches and subsequently merges them to obtain global information.

By integrating the information from both branches, TPGN achieves comprehensive semantic information capture for accurate predictions. It also should be noted that other methods can substitute PGN and be used in the long-term information extraction branch of TPGN, undoubtedly enabling TPGN to be a general framework to model temporal dependencies. Furthermore, TPGN maintains a efficient computational complexity of $\mathcal{O}(\sqrt{L})$. To better illustrate the advantages of TPGN, inspired by [Jia et al., 2023], we have provided a information propagation comparative diagram in Figure 1 and an analysis table in Table 3.

The main contributions of this paper can be summarized as follows:

- We propose a novel general paradigm called PGN as the new successor to RNN, as shown in Figure 1 (l). It reduces the information propagation path to $\mathcal{O}(1)$, enabling better capture of long-term dependencies in input signals and addressing the limitations of RNNs.

- We propose TPGN, a novel temporal modeling framework based on PGN, which comprehensively captures semantic information through two branches, as shown in Figure 1 (m). One branch utilizes PGN to capture long-term periodic patterns and preserve their local characteristics, while the other branch employs patches to capture short-term information and aggregates them to obtain a global representation of the series. Notably, TPGN can also accommodate other models, making it be a general temporal modeling framework.

- In terms of efficiency, PGN maintains the same complexity of $\mathcal{O}(L)$ as RNN. However, due to its parallelizable calculations, PGN achieves higher actual efficiency. On the other hand, TPGN, serving as a general temporal modeling framework, exhibits a favorable complexity of $\mathcal{O}(\sqrt{L})$. For a more detailed comparison of complexities, please refer to Table 3.

- We conducted experiments on five benchmark datasets, and the results indicated that TPGN achieved an average MSE improvement of 12.35% in various long-range time series forecasting tasks compared to the previous best-performing models. Furthermore, in comparison to the average performance of specific models across all tasks, TPGN achieved an average MSE improvement ranging from 14.08% to 39.65%. Additionally, experimental evaluations on computational complexity confirmed the efficiency of TPGN.

## 2 Related Works

### 2.1 Modeling Interaction Cross Temporal Dimension

The methods that focus on temporal modeling can be broadly categorized into four paradigms: RNN-, CNN-, MLP- (Linear-), and Transformer-based. The limitations of RNNs [Hochreiter and Schmidhuber, 1997, Chung et al., 2014, Salinas et al., 2020] have been discussed in Section 1. Despite some methods [Chang et al., 2017, Yu and Liu, 2018, Jia et al., 2023] trying to alleviate these limitations, the recurrent structure still hinders their further development. CNNs [Franceschi et al., 2019, Sen et al., 2019] offer advantages in efficiency and shorter information propagation paths, but primarily constrained by limited receptive fields [Wu et al., 2023], resulting in an increase in the information propagation path as the length of the processed signal increases. Although some methods [Wang et al., 2023, Luo and Wang, 2024] have increased the receptive field to address these issues, the 1D modeling approach makes it challenging for them to directly capture periodicity. The advantages of Linear [Zeng et al., 2023] lie in its simple structure and high operational efficiency. Some advanced models have further enhanced the performance of MLP or Linear by applying them in the frequency domain [Xu et al., 2024] or introducing multi scales [Wang et al., 2024], which could lead to higher execution overhead. Classic Transformer-based methods either struggle to capture semantic information [Wu et al., 2023, Nie et al., 2023] due to point-wise attention mechanisms [Vaswani et al., 2017, Zhou et al., 2021, 2022a] or have high complexity [Wu et al., 2021, Liu et al., 2022a], limiting their ability. Subsequently, this problem was effectively addressed by

utilizing patches [Nie et al., 2023]. However, they still suffer from the 1D modeling issue mentioned earlier or the problem of limited receptive fields. More detailed discussion and analysis can be found in Appendix B.

## 2.2 Modeling Interaction Cross Variable Dimension

For handling variable dimensions, there are generally four categories: variable fusion processing, variable independent processing, modeling based on Transformers, and modeling based on Graph Neural Networks (GNNs). Traditional fusion processing methods, due to the heterogeneity of multiple variables [Zhou et al., 2021], introduce excessive noise, resulting in worse performance compared to independent processing of variables [Nie et al., 2023]. However, by applying attention mechanisms and Graph Neural Networks (GNN) on the variable dimension to replace independent modeling of variables, it is possible to successfully capture the correlations and differences between variables, thereby significantly improving the performance of multivariate modeling. Representative methods for modeling variable relationships based on Transformers include Crossformer [Zhang and Yan, 2022] and iTransformer [Liu et al., 2024], while GNN-based representative methods include CrossGNN [Huang et al., 2023] and FourierGNN [Yi et al., 2023]. They provide excellent inspiration for analyzing and modeling multivariate time series.

## 3 Methodology

In this section, we first introduce our proposed novel paradigm called **P**arallel **G**ated **N**etwork (**PGN**), and explain how it reduces the information propagation paths, overcomes the limitations of RNNs, and emerges as the new successor to RNNs. Next, we present our newly designed temporal modeling framework called **T**emporal **PGN** (**TPGN**), which incorporates two separate branches to comprehensively capture semantic information. Finally, we provide a comprehensive complexity analysis to evaluate the computational efficiency of our methods.

### 3.1 Parallel Gated Network (PGN)

Building upon the previous analysis, the limitation of RNNs lies in the excessively long information propagation paths of its recurrent structure, which directly leads to a series of issues, such as difficulty in capturing long-term dependencies (performance), low efficiency in sequential computations (efficiency), and gradient exploding/vanishing (training difficulty). Indeed, some RNNs leverage specialized gated mechanisms, such as LSTM [Hochreiter and Schmidhuber, 1997] and GRU [Chung et al., 2014], which do have advantages in information selection and fusion. However, when faced with the disastrous limitation of RNNs, their advantages become insignificant.

Based on this, we propose a novel general paradigm called PGN as the new successor to RNNs. PGN draws the advantages of RNNs while reducing information propagation paths to $\mathcal{O}(1)$, thereby addressing the limitation of RNNs. The information propagation illustration and structure of PGN are shown in Figure 1 (l) and Figure 2 (a), respectively. On one hand, to enable PGN to capture information from all preceding time steps within short information propagation paths, we introduce a linear **H**istorical **I**nformation **E**xtraction (HIE) layer to aggregate information from the entire history at each time step. Importantly, at this stage, the computation of each time step of the signal is independent of others, allowing for effective parallel processing. On the other hand, PGN leverages gated mechanisms to inherit the advantages of information selection and fusion. It is important to emphasize that in PGN, we utilize only a single gate to simultaneously control the information selection and fusion in a parallel manner across all time steps of the sequence, resulting in reduced computational overhead. When given an input signal $X \in \mathbb{R}^L$ of length $L$, the computation process of PGN can be formalized as follows:

$$
\begin{aligned}
H &= \text{HIE}(\text{Padding}(X)), \\
G &= \sigma(W_g[X, \ H] + b_g), \\
\hat{H} &= \tanh(W_t[X, \ H] + b_t), \\
Out &= G \odot H + (1 - G) \odot \hat{H},
\end{aligned}
\tag{1}
$$

where Padding($\cdot$) represents the operation of filling the front of the processed signal along the length dimension with a zero-filled vector of size $\mathbb{R}^{(L-1)}$. HIE($\cdot$) is a linear layer with weight matrices

$W_\text{h} \in \mathbb{R}^{d_\text{m} \times (L-1)}$ and bias vectors $b_h \in \mathbb{R}^{d_\text{m}}$. It aggregates all relevant historical information for each time step in parallel by sliding along the sequence length dimension, and $H \in \mathbb{R}^{L \times d_\text{m}}$ represents the output of this operation. The weight matrices $W_g, W_t \in \mathbb{R}^{d_\text{m} \times (d_\text{m}+1)}$ and bias vectors $b_g, b_t \in \mathbb{R}^{d_\text{m}}$ are utilized in the computations. $G$ and $\hat{H}$ are the intermediate variables involved in the gated mechanism. The symbol $\odot$ represents the element-wise product, while $\sigma(\cdot)$ and $\tanh(\cdot)$ denote the sigmoid and tanh activation functions. $Out \in \mathbb{R}^{L \times d_\text{m}}$ represents the output of PGN.

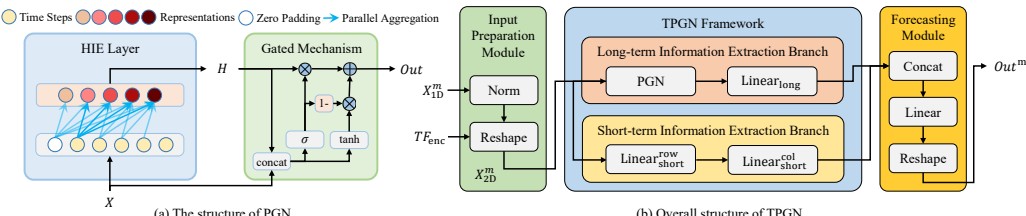

Figure 2: The structures of PGN and TPGN.

## 3.2 Temporal Parallel Gated Network (TPGN)

The specific objective of time series forecasting task is to predict the future series of length $L_\text{f}$ given a historical sequence of length $L_\text{h}$. As stated in Section 1, PGN may not be effective in directly extracting periodic semantic information, which limits its application in time series forecasting tasks. Inspired by [Wu et al., 2023, Jia et al., 2023, Dai et al., 2024], we transform the input series from 1D to 2D for modeling. To fully capture the short-term changes and long-term periodic patterns with different characteristics in the rows and columns of the 2D input, we introduce two branches to model them separately. The information propagation diagram and overall structure of TPGN are shown in Figure 1 (m) and Figure 2 (b).

**Input Preparation Module** To enable TPGN to directly capture periodic semantic information, inspired by previous works [Wu et al., 2023, Jia et al., 2023, Dai et al., 2024], we reshape the series from 1D to 2D. Notably, we do not need to introduce multiple scales of periods like in TimesNet [Wu et al., 2023] and PDF [Dai et al., 2024], as it would result in increased computational overhead. Instead, we draw inspiration from WITRAN [Jia et al., 2023] and solely reset the sequence based on the natural scale of the time series. In addition, to minimize the negative impact of data fluctuations on model training, inspired by [Liu et al., 2022b, 2024], we have introduced a normalization layer along the temporal dimension. When given an input sequence $X_\text{1D} = \{x_1, x_2, \ldots, x_{L_\text{h}}\} \in \mathbb{R}^{L_\text{h} \times C}$ and temporal external feature $TF_\text{enc} \in \mathbb{R}^{L_\text{h} \times C_\text{time}}$ ($C$ and $C_\text{time}$ represent the number of variables and temporal external features), this module can be mathematically expressed as:

$$
\mu_X = \frac{1}{L_\text{h}} \sum_{i=1}^{L_\text{h}} x_i, \ \sigma_X^2 = \frac{1}{L_\text{h}} \sum_{i=1}^{L_\text{h}} (x_i - \mu_X)^2,
$$
$$
X_\text{1D}^{norm} = \begin{cases} X_\text{1D}, & norm = 0 \\ (X_\text{1D} - \mu_X)/\sigma_X, & norm = 1 \end{cases},
$$
$$
X_\text{2D} = \text{Reshape}([X_\text{1D}^{norm}, \ TF_\text{enc}]).
$$

(2)

Here, $X_\text{1D}^{norm} \in \mathbb{R}^{L_\text{h} \times C}$ represent the normalized series, $[\cdot]$ represents the concat operation, and the hyperparameters $norm$ should be determined based on the characteristics of different datasets. To combine each variable of the input series with the temporal feature, we need to expand $X_\text{1D}^{norm}$ by adding an extra dimension to match the shape of $\mathbb{R}^{L_\text{h} \times C \times 1}$. Additionally, $TF_\text{enc}$ needs to be expanded by adding a dimension and repeated $C$ times to match the shape of $\mathbb{R}^{L_\text{h} \times C \times C_\text{time}}$. Afterwards, we concatenate the results and reshape them according to the natural period $P$ of series through $\text{Reshape}(\cdot)$, and $X_\text{2D} \in \mathbb{R}^{R \times P \times C \times (1+C_\text{time})}$ represents the output of this module. $R$ and $P$ represent the number of rows and columns in the 2D input, respectively.

**TPGN** As TPGN focuses on modeling the temporal dimension, which is crucial for any variable in the time series. In the following discussions, we will focus on an example variable $m$ to provide

a detailed explanation, where $X_{2D}^m \in \mathbb{R}^{R \times P \times (1+C_{\text{time}})}$ represents the input. To better capture long- and short- term information while preserving their respective characteristics, we have designed two branches as illustrated in Figure 1 (m) and Figure 2 (b).

**In the long-term information extraction branch**, we directly capture the information using PGN. On one hand, it effectively captures the long-term repetitive historical information for each time step. On the other hand, through the gated mechanism, it selects and fuses the current and historical information at each time step, thereby preserving the long-term periodic characteristics to the maximum extent. Specifically, this branch can be formulated as follows:

$$X_{\text{long}}^m = \text{PGN}(X_{2D}^m), \quad H_{\text{long}}^m = \text{Linear}_{\text{long}}(X_{\text{long}}^m), \tag{3}$$

where $\text{PGN}(\cdot)$ represents the input being passed through the PGN paradigm. It is important to note that PGN operates along the $R$ dimension. The advantage of this approach is it preserves the individual characteristics of each column, better serving forecasting. The output is denoted as $X_{\text{long}}^m \in \mathbb{R}^{R \times P \times d_m}$. To facilitate the utilization of long-term information for prediction purposes, we aggregate the information from all rows in each column using a linear layer $\text{Linear}_{\text{long}}(\cdot)$. The output of this branch is denoted as $H_{\text{long}}^m \in \mathbb{R}^{P \times d_m}$.

**In the short-term information extraction branch**, considering the advantage of patch in aggregation short-term information, we first utilize a linear layer to aggregate the short-term information into patches. Then, another linear layer is used to further fuse the patches into the global information of the series. The specific process can be formulated as follows:

$$H_{\text{short}}^m = \text{Linear}_{\text{short}}^{\text{row}}(X_{2D}^m), \quad H_{\text{global}}^m = \text{Linear}_{\text{short}}^{\text{col}}(H_{\text{short}}^m), \tag{4}$$

where $\text{Linear}_{\text{short}}^{\text{row}}(\cdot)$ operates along the $P$ dimension, and $H_{\text{short}}^m \in \mathbb{R}^{R \times d_m}$ is its output. Subsequently, $\text{Linear}_{\text{short}}^{\text{col}}(\cdot)$ further aggregates the patches $H_{\text{short}}^m$ and obtains the global representation $H_{\text{global}}^m \in \mathbb{R}^{1 \times d_m}$ of the sequence. Finally, to facilitate subsequent predictions, we repeat $H_{\text{global}}^m$ along the first dimension $P$ times to obtain a new representation with the same dimension $\mathbb{R}^{P \times d_m}$ as the output of the long-term information extraction branch.

**Forecasting Module**    In this module, begin by concatenating the information representations derived from the two branches of TPGN. The concatenated information encompasses the local long-term periodic characteristics observed across various columns in the 2D input series, along with the globally aggregated short-term information. Subsequently, we take the previously aggregated representation with comprehensive semantic information and pass it through a linear layer to predict future values at different positions within the period. The formulation of this module is as follows:

$$Out^m = \text{Reshape}(\text{Linear}([H_{\text{long}}^m, \ H_{\text{global}}^m])), \tag{5}$$

where $[\cdot]$ represents the concat operation. The output dimension after $\text{Linear}(\cdot)$ is $\mathbb{R}^{P \times R_f}$, where $R_f$ multiplied by $P$ equal forecasting series length $L_f$. Finally, the above output will be permuted and reshaped to 1D dimension by $\text{Reshape}(\cdot)$ operation, the result $Out^m \in \mathbb{R}^{L_f}$.

### 3.3    Complexity Analysis

**PGN**    While PGN processes signals in the time dimension in parallel, each step still involves processing all its historical information. Therefore, the theoretical complexity of this part is still linear with respect to the length $L$ of the signal being processed. The complexity of the gated mechanism is independent of the signal length, so the complexity of PGN can be expressed as $\mathcal{O}(L)$. Noted that PGN has the same theoretical complexity as RNN, but due to the parallelized ability of PGN computations, it has much higher efficiency in practice compared to RNN.

**TPGN**    Since TPGN has two separate branches, it is necessary to analyze them separately.

For the long-term information extraction branch, TPGN applies the PGN paradigm along the number of $R$, the complexity of this step is indeed linear with respect to $R$, denoted as $\mathcal{O}(R)$. The aggregation of all rows of information is accomplished through a linear layer, which still has a complexity proportional to $\mathcal{O}(R)$. Therefore, the complexity of the long-term information extraction branch can be expressed as $\mathcal{O}(R)$.

For the short-term information extraction branch, TPGN applies two linear layers. The first linear layer compresses the time dimension from $P$ to 1, while the second linear layer compresses the other time dimension $R$ to 1, therefore, their complexities are respectively $\mathcal{O}(P)$ and $\mathcal{O}(R)$.

Since $R$ multiplied by $P$ equals the input sequence length $L_{\mathrm{h}}$ ($L$), the complexities $\mathcal{O}(R)$ and $\mathcal{O}(P)$ are both equal to $\mathcal{O}(\sqrt{L})$. For the two branches of TPGN, the complexities are both $\mathcal{O}(\sqrt{L})$. Therefore, the complexity of TPGN is also $\mathcal{O}(\sqrt{L})$.

## 4 Experiments

**Datasets** To validate the performance of TPGN, we followed WITRAN [Jia et al., 2023] and conducted experiments on five real-world benchmark datasets that span across energy, traffic, and weather domains. More details about the datasets can be found in Appendix C.

**Baselines** We conducted a comprehensive comparison of thirteen methods in our study. These methods include two RNN-based methods: WITRAN [Jia et al., 2023], SegRNN [Lin et al., 2023], three CNN-based methods: ModernTCN [Luo and Wang, 2024], TimesNet [Wu et al., 2023], MICN [Wang et al., 2023], three MLP-based methods: FITS [Xu et al., 2024], TimeMixer [Wang et al., 2024], DLinear [Zeng et al., 2023], four Transformer-based methods: iTransformer [Liu et al., 2024], PDF [Dai et al., 2024], Basisformer [Ni et al., 2023], PatchTST [Nie et al., 2023], and FiLM [Zhou et al., 2022b]. It should be noted that certain earlier methods such as Vanilla-Transformer [Vaswani et al., 2017], Informer [Zhou et al., 2021], Autoformer [Wu et al., 2021], Pyraformer [Liu et al., 2022a], and FEDformer [Zhou et al., 2022a] have been extensively surpassed by the methods we selected. Hence, we did not include these earlier methods as baselines in our comparison. For further discussion on these methods and details of the experimental setup, please refer to Appendix B and Appendix D.

### 4.1 Experimental Results

It is important to emphasize that while there have been numerous works focusing on modeling the relationships among multiple variables in time series, they still need to effectively capture information along the temporal dimension to better accommodate multivariate time series. In contrast, our method primarily concentrates on modeling the temporal dimension. To mitigate any potential negative impact caused by the heterogeneity of multivariate data, we followed the experimental setup of WITRAN [Jia et al., 2023], conducted experiments using a single variable. To ensure fairness, we conducted parameter search for each baseline model, enabling them to achieve their respective optimal performance across different tasks. For further experiment details, please refer to Appendix D.

Table 1: **Long-range** forecasting results. A lower MSE or MAE indicates a better prediction. The best results are highlighted in **bold** and the second best results are underlined.

| Methods | | TPGN (ours) | | WITRAN | | SegRNN | | ModernTCN | | TimesNet | | MICN | | FITS | | TimeMixer | | DLinear | | iTransformer | | PDF | | Basisformer | | PatchTST | | FiLM | |
|---|---|---|---|---|---|---|---|---|---|---|---|---|---|---|---|---|---|---|---|---|---|---|---|---|---|---|---|---|---|
| Metric | | MSE | MAE | MSE | MAE | MSE | MAE | MSE | MAE | MSE | MAE | MSE | MAE | MSE | MAE | MSE | MAE | MSE | MAE | MSE | MAE | MSE | MAE | MSE | MAE | MSE | MAE | MSE | MAE |
| ECL | 168-168 | **0.2107** | **0.3264** | 0.2397 | 0.3519 | 0.2600 | 0.3622 | 0.2473 | 0.3437 | 0.2825 | 0.3797 | 0.3168 | 0.3797 | 0.2598 | 0.3573 | 0.2804 | 0.3792 | 0.2606 | 0.3579 | 0.2479 | 0.3516 | 0.2483 | 0.3491 | 0.3116 | 0.4026 | 0.2980 | 0.3832 | 0.2587 | 0.3557 |
| | 168-336 | **0.2276** | **0.3446** | 0.2607 | 0.3721 | 0.3166 | 0.4017 | 0.3110 | 0.3887 | 0.3505 | 0.4253 | 0.3002 | 0.4253 | 0.3072 | 0.3938 | 0.3183 | 0.4029 | 0.3080 | 0.3946 | 0.3128 | 0.3974 | 0.3094 | 0.3902 | 0.4844 | 0.4824 | 0.3446 | 0.4094 | 0.3062 | 0.3922 |
| | 168-720 | **0.2303** | **0.3550** | 0.2906 | 0.3965 | 0.3964 | 0.4660 | 0.3624 | 0.4478 | 0.4261 | 0.4686 | 0.4453 | 0.4686 | 0.3504 | 0.4366 | 0.3835 | 0.4560 | 0.3515 | 0.4374 | 0.3660 | 0.4438 | 0.3541 | 0.4423 | 0.6448 | 0.5653 | 0.4324 | 0.4782 | 0.3486 | 0.4349 |
| | 168-1440 | **0.2484** | **0.3775** | 0.3255 | 0.4302 | 0.7574 | 0.6547 | 0.5307 | 0.5573 | 0.6688 | 0.6102 | 0.8784 | 0.6102 | 0.5176 | 0.5591 | 0.6857 | 0.6194 | 0.5300 | 0.5681 | 0.7028 | 0.6348 | 0.9029 | 0.6913 | 0.6368 | 0.5967 | 0.7349 | 0.6464 | 0.5146 | 0.5565 |
| Traffic | 168-168 | **0.1196** | **0.1857** | 0.1377 | 0.2051 | 0.1900 | 0.2816 | 0.1473 | 0.2212 | 0.1490 | 0.2293 | 0.2418 | 0.3537 | 0.1498 | 0.2134 | 0.1340 | 0.2124 | 0.1519 | 0.2195 | 0.1343 | 0.2083 | 0.1397 | 0.2119 | 0.1634 | 0.2553 | 0.1622 | 0.2330 | 0.1501 | 0.2143 |
| | 168-336 | **0.1156** | **0.1868** | 0.1321 | 0.2059 | 0.2227 | 0.3129 | 0.1410 | 0.2214 | 0.1499 | 0.2356 | 0.2420 | 0.3568 | 0.1445 | 0.2148 | 0.1298 | 0.2147 | 0.1468 | 0.2210 | 0.1366 | 0.2221 | 0.1351 | 0.2132 | 0.1544 | 0.2493 | 0.1641 | 0.2364 | 0.1453 | 0.2165 |
| | 168-720 | **0.1293** | **0.2057** | 0.1439 | 0.2226 | 0.2674 | 0.3436 | 0.1574 | 0.2389 | 0.1621 | 0.2471 | 0.2488 | 0.3592 | 0.1603 | 0.2330 | 0.1396 | 0.2285 | 0.1629 | 0.2389 | 0.1402 | 0.2265 | 0.1502 | 0.2290 | 0.1538 | 0.2490 | 0.1770 | 0.2548 | 0.1617 | 0.2358 |
| | 168-1440 | **0.1390** | **0.2114** | 0.1611 | 0.2369 | 0.3453 | 0.3929 | 0.1980 | 0.2739 | 0.1691 | 0.2517 | 0.2817 | 0.3818 | 0.1845 | 0.2571 | 0.1547 | 0.2392 | 0.1890 | 0.2640 | 0.1519 | 0.2321 | 0.2074 | 0.2779 | 0.1735 | 0.2654 | 0.2139 | 0.2875 | 0.1861 | 0.2615 |
| ETTh₁ | 168-168 | **0.1061** | **0.2533** | 0.1105 | 0.2589 | 0.1189 | 0.2705 | 0.1210 | 0.2694 | 0.1133 | 0.2612 | 0.1257 | 0.2803 | 0.1089 | 0.2556 | 0.1122 | 0.2605 | 0.1112 | 0.2598 | 0.1115 | 0.2579 | 0.1169 | 0.2646 | 0.1212 | 0.2704 | 0.1091 | 0.2558 |
| | 168-336 | **0.1110** | **0.2625** | 0.1189 | 0.2714 | 0.1378 | 0.2972 | 0.1342 | 0.2884 | 0.1202 | 0.2732 | 0.1422 | 0.3006 | 0.1162 | 0.2682 | 0.1209 | 0.2716 | 0.1251 | 0.2794 | 0.1203 | 0.2709 | 0.1207 | 0.2725 | 0.1227 | 0.2734 | 0.1287 | 0.2808 | 0.1187 | 0.2708 |
| | 168-720 | **0.1346** | **0.2908** | 0.1566 | 0.3150 | 0.2134 | 0.3697 | 0.1676 | 0.3038 | 0.1458 | 0.3059 | 0.1609 | 0.3200 | 0.1362 | 0.2927 | 0.1544 | 0.3109 | 0.1919 | 0.3465 | 0.1423 | 0.3020 | 0.1720 | 0.3278 | 0.1521 | 0.3121 | 0.1727 | 0.3297 | 0.1717 | 0.3266 |
| | 168-1440 | **0.1343** | **0.2941** | 0.1541 | 0.3157 | 0.4033 | 0.5296 | 0.2756 | 0.4247 | 0.1543 | 0.3119 | 0.1444 | 0.3032 | 0.2319 | 0.3863 | 0.1480 | 0.3068 | 0.3606 | 0.4939 | 0.1520 | 0.3107 | 0.2792 | 0.4272 | 0.1664 | 0.3230 | 0.3206 | 0.4561 | 0.3056 | 0.4494 |
| ETTh₂ | 168-168 | **0.2174** | **0.3623** | 0.2389 | 0.3813 | 0.2566 | 0.4013 | 0.2564 | 0.3980 | 0.2655 | 0.4051 | 0.2734 | 0.4162 | 0.2547 | 0.3947 | 0.2507 | 0.3936 | 0.2556 | 0.3944 | 0.2630 | 0.4053 | 0.2606 | 0.4012 | 0.2806 | 0.4138 | 0.2582 | 0.3983 | 0.2546 | 0.3942 |
| | 168-336 | **0.2237** | **0.3769** | 0.2277 | 0.3778 | 0.3017 | 0.4398 | 0.2918 | 0.4312 | 0.2725 | 0.4163 | 0.3017 | 0.4429 | 0.2642 | 0.4085 | 0.2642 | 0.4085 | 0.2891 | 0.4256 | 0.2658 | 0.4092 | 0.3064 | 0.4440 | 0.2697 | 0.4132 | 0.3206 | 0.4514 | 0.2894 | 0.4263 |
| | 168-720 | **0.2356** | **0.3898** | 0.2718 | 0.4146 | 0.3897 | 0.4988 | 0.3991 | 0.5022 | 0.3186 | 0.4465 | 0.4770 | 0.5602 | 0.3983 | 0.5023 | 0.3259 | 0.4510 | 0.4090 | 0.5090 | 0.2951 | 0.4312 | 0.4546 | 0.5415 | 0.3034 | 0.4372 | 0.4398 | 0.5304 | 0.4039 | 0.5061 |
| | 168-1440 | **0.2514** | **0.4070** | 0.3350 | 0.4624 | 0.8067 | 0.7307 | 0.8537 | 0.7437 | 0.3839 | 0.4933 | 0.4876 | 0.5602 | 0.3839 | 0.4933 | 0.7852 | 0.7256 | 0.3937 | 0.5040 | 0.7921 | 0.7262 | 0.3806 | 0.4894 | 0.8699 | 0.7651 | 0.3963 | 0.4996 | 0.8339 | 0.7330 | 0.7843 | 0.7210 |
| Weather | 168-168 | **0.1877** | **0.3166** | 0.2050 | 0.3338 | 0.2165 | 0.3405 | 0.2692 | 0.4088 | 0.2420 | 0.3608 | 0.2231 | 0.3489 | 0.2423 | 0.3561 | 0.2275 | 0.3466 | 0.2421 | 0.3578 | 0.2636 | 0.4056 | 0.2557 | 0.3989 | 0.2301 | 0.3541 | 0.2469 | 0.3597 | 0.2426 | 0.3544 |
| | 168-336 | **0.1978** | **0.3278** | 0.2197 | 0.3470 | 0.2916 | 0.4314 | 0.2821 | 0.3885 | 0.2663 | 0.3837 | 0.2977 | 0.4007 | 0.2775 | 0.3836 | 0.2918 | 0.3975 | 0.2658 | 0.4092 | 0.3065 | 0.4404 | 0.2593 | 0.3751 | 0.3040 | 0.4049 | 0.2981 | 0.3988 |
| | 168-720 | **0.1925** | **0.3255** | 0.2538 | 0.3796 | 0.3478 | 0.4519 | 0.4279 | 0.4812 | 0.2941 | 0.4013 | 0.3077 | 0.4202 | 0.4044 | 0.4758 | 0.2873 | 0.3931 | 0.3915 | 0.4739 | 0.2754 | 0.3900 | 0.3988 | 0.4673 | 0.2834 | 0.3916 | 0.4023 | 0.4662 | 0.4093 | 0.4737 |
| | 168-1440 | **0.1786** | **0.3184** | 0.2695 | 0.3966 | 0.6456 | 0.6484 | 0.7049 | 0.6351 | 0.2988 | 0.4092 | 0.4306 | 0.4994 | 0.7027 | 0.6555 | 0.3010 | 0.4078 | 0.5837 | 0.6177 | 0.3007 | 0.4113 | 0.7334 | 0.6557 | 0.2959 | 0.4095 | 0.7150 | 0.6336 | 0.7028 | 0.6453 |
| **Average improvement of TPGN** | | | | 14.08% | 8.12% | 39.65% | 26.31% | 32.92% | 20.58% | 25.42% | 15.23% | 36.15% | 24.03% | 30.30% | 17.97% | 21.58% | 12.89% | 32.75% | 19.99% | 21.40% | 13.32% | 33.17% | 20.92% | 28.01% | 17.42% | 37.44% | 22.62% | 31.68% | 18.81% |

**Long-range Forecasting Results** We conducted four tasks on each dataset for long-range forecasting, and the results are shown in Table 1. For instance, considering the task setting 168-1440 on the left side of Table 1, it signifies that the input length is 168, and the forecasting length is 1440. It is worth noting that our proposed TPGN achieves state-of-the-art (SOTA) performance across all tasks, with an average improvement of MSE by 12.35% and MAE by 7.25% compared to the previous best methods. In particular, TPGN demonstrates an average reduction in MSE of 17.31% for the ECL dataset, 9.38% for the Traffic dataset, 3.79% for the ETTh₁ dataset, 12.26% for the ETTh₂ dataset,

and 19.09% for the Weather dataset. Furthermore, we calculated the average improvement values of TPGN compared to each method across all tasks and displayed them in the last row of Table 1. Based on the aforementioned results, it can be concluded that TPGN is capable of effectively handling long-range forecasting tasks in various domains. For further experimental results and showcases, please refer to Appendix E and Appendix I.

**Performance of Variations with Different Forecasting Lengths**    It is important to emphasize that through Table 1, we observed that as the forecasting task length increased, all models generally experienced varying degrees of performance decline. However, TPGN appeared to exhibit slower decline trend. To further validate the performance of TPGN, we expanded our experimental settings by selecting representative methods from different paradigms in Table 1: WITRAN (RNN-Based), TimesNet (CNN-Based), TimeMixer (MLP-Based), and iTransformer (Transformer-Based), and compared them with TPGN. The experimental results on the ECL dataset are depicted in Figure 3, and more experimental results can be found in Appendix F. It can be observed that as the forecasting task length gradually increases, TPGN exhibits a stable decline in performance and consistently outperforms the other comparative methods. This strongly indicates that TPGN effectively captures the comprehensive information contained in fewer inputs and applies it well to the forecasting tasks.

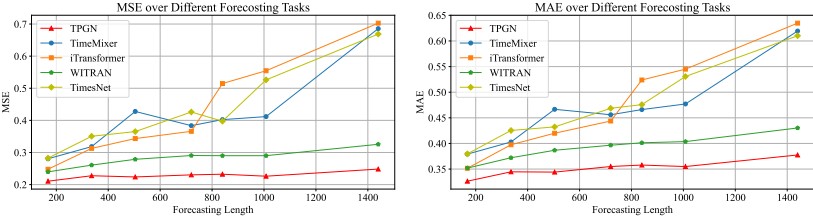

Figure 3: Experimental results with different forecasting lengths on the ECL dataset.

For other aspects experiments and analysis of our methods can be found in Appendix G.

## 4.2 Ablation Study

To validate the roles of the two information extraction branches in TPGN, we conducted tests on the performance of the model when using only one branch. Additionally, to verify the effectiveness of PGN, we performed ablation experiments by replacing PGN with GRU and MLP, respectively. "TPGN-long" represents only using the long-term information extraction branch, while "TPGN-short" represents only using the short-term. "TPGN-GRU/-LSTM/-MLP/-Attn" respectively represent replacing PGN in the long-term information extraction branch with GRU, LSTM, MLP and self-attention. The results of these experiments are presented in Table 2.

Table 2: Results of the **ablation study** on long-range forecasting tasks. A lower MSE or MAE indicates a better prediction. The best results are highlighted in **bold**.

| | Methods | **TPGN** | | TPGN-long | | TPGN-short | | TPGN-GRU | | TPGN-LSTM | | TPGN-MLP | | TPGN-Attn | |
| | Metric | MSE | MAE | MSE | MAE | MSE | MAE | MSE | MAE | MSE | MAE | MSE | MAE | MSE | MAE |
|---|---|---|---|---|---|---|---|---|---|---|---|---|---|---|---|
| ECL | 168-168 | **0.2107** | **0.3264** | 0.2223 | 0.3399 | 0.7226 | 0.6755 | 0.2363 | 0.3388 | 0.2263 | 0.3344 | 0.2377 | 0.3425 | 0.2279 | 0.3376 |
| | 168-336 | **0.2276** | **0.3446** | 0.2422 | 0.3567 | 0.7598 | 0.6901 | 0.2393 | 0.3497 | 0.2591 | 0.3641 | 0.2669 | 0.3636 | 0.2660 | 0.3709 |
| | 168-720 | **0.2303** | **0.3550** | 0.2405 | 0.3628 | 0.7841 | 0.6997 | 0.2719 | 0.3774 | 0.2703 | 0.3855 | 0.2967 | 0.3946 | 0.3149 | 0.4115 |
| | 168-1440 | **0.2484** | **0.3775** | 0.2710 | 0.3951 | 0.8323 | 0.7224 | 0.3557 | 0.4550 | 0.2936 | 0.4197 | 0.3456 | 0.4424 | 0.3649 | 0.4589 |
| Traffic | 168-168 | **0.1196** | **0.1857** | 0.1215 | 0.1871 | 1.8730 | 1.1806 | 0.1269 | 0.1923 | 0.1271 | 0.1920 | 0.1456 | 0.2139 | 0.1391 | 0.2145 |
| | 168-336 | **0.1156** | 0.1868 | 0.1166 | **0.1867** | 1.8665 | 1.1790 | 0.1204 | 0.1892 | 0.1195 | 0.1926 | 0.1419 | 0.2174 | 0.1378 | 0.2209 |
| | 168-720 | **0.1293** | 0.2057 | 0.1294 | **0.2041** | 1.8548 | 1.1746 | 0.1306 | 0.2063 | 0.1307 | 0.2109 | 0.1565 | 0.2349 | 0.1565 | 0.2436 |
| | 168-1440 | **0.1390** | **0.2114** | 0.1391 | 0.2119 | 1.8589 | 1.1721 | 0.1440 | 0.2168 | 0.1435 | 0.2157 | 0.1838 | 0.2567 | 0.1987 | 0.2838 |
| ETTh₁ | 168-168 | **0.1061** | **0.2533** | 0.1153 | 0.2666 | 0.1101 | 0.2594 | 0.1081 | 0.2548 | 0.1090 | 0.2560 | 0.1079 | 0.2549 | 0.1092 | 0.2569 |
| | 168-336 | **0.1110** | **0.2625** | 0.1163 | 0.2698 | 0.1183 | 0.2729 | 0.1117 | 0.2641 | 0.1120 | 0.2652 | 0.1117 | 0.2652 | 0.1134 | 0.2636 |
| | 168-720 | **0.1346** | **0.2908** | 0.1399 | 0.2971 | 0.1416 | 0.2994 | 0.1462 | 0.3057 | 0.1464 | 0.3068 | 0.1356 | 0.2930 | 0.1425 | 0.2995 |
| | 168-1440 | **0.1343** | **0.2941** | 0.1352 | 0.2949 | 0.1551 | 0.3115 | 0.1497 | 0.3079 | 0.1502 | 0.3091 | 0.1544 | 0.3112 | 0.1572 | 0.3141 |
| ETTh₂ | 168-168 | **0.2174** | **0.3623** | 0.2402 | 0.3850 | 0.3250 | 0.4531 | 0.2572 | 0.3969 | 0.2567 | 0.3918 | 0.2472 | 0.3906 | 0.2509 | 0.3960 |
| | 168-336 | **0.2237** | **0.3769** | 0.2477 | 0.3969 | 0.3312 | 0.4535 | 0.2587 | 0.4014 | 0.2647 | 0.4141 | 0.2550 | 0.3987 | 0.2652 | 0.4072 |
| | 168-720 | **0.2356** | **0.3898** | 0.2475 | 0.3995 | 0.3382 | 0.4617 | 0.2619 | 0.4072 | 0.2714 | 0.4101 | 0.2698 | 0.4144 | 0.2744 | 0.4203 |
| | 168-1440 | **0.2514** | **0.4070** | 0.2932 | 0.4341 | 0.3723 | 0.4841 | 0.2611 | 0.4136 | 0.2865 | 0.4225 | 0.2901 | 0.4358 | 0.4226 | 0.5147 |
| Weather | 168-168 | **0.1877** | **0.3166** | 0.2035 | 0.3317 | 0.2690 | 0.3864 | 0.2184 | 0.3445 | 0.2024 | 0.3323 | 0.2354 | 0.3555 | 0.2269 | 0.3594 |
| | 168-336 | **0.1978** | **0.3278** | 0.2088 | 0.3401 | 0.3138 | 0.4215 | 0.2222 | 0.3540 | 0.2182 | 0.3476 | 0.2790 | 0.3919 | 0.2517 | 0.3820 |
| | 168-720 | **0.1925** | **0.3255** | 0.2028 | 0.3351 | 0.3576 | 0.4573 | 0.2139 | 0.3479 | 0.2060 | 0.3392 | 0.3226 | 0.4319 | 0.2796 | 0.4044 |
| | 168-1440 | **0.1786** | **0.3184** | 0.1823 | 0.3218 | 0.4523 | 0.5372 | 0.1969 | 0.3309 | 0.1896 | 0.3244 | 0.4198 | 0.5164 | 0.3417 | 0.4639 |

Through the ablation experiments, we can draw the following conclusions: (1) The two branches designed in TPGN are reasonable as they capture long-term and short-term information respectively, while preserving their respective characteristics. In most cases, using only one branch leads to subpar results due to incomplete capture of essential features. (2) The branch capturing long-term information

in TPGN is more significant. This can be observed by comparing the performance degradation when using only one branch versus using both branches together. Especially for strongly periodic data like traffic, in some tasks, using only the long-term information capture branch can achieve good results. This also aligns with our earlier mention in Section 1 about the significance of prioritizing the modeling of periodicity. (3) Compared to GRU and LSTM, which have more gates, PGN introduces only one gate but still achieves better performance. This strongly demonstrates the ability of PGN to serve as the new successor to RNN. (4) The comparison between "TPGN-GRU/-LSTM/-MLP/-Attn" and the baseline results demonstrates the strong generality and performance of the TPGN framework. Despite their inferior performance compared to TPGN, in some tasks, they even surpass the previous SOTA time series forecasting methods.

## 4.3 Efficiency of Execution

Although this paper primarily focuses on predicting longer-range future outputs using short-range historical inputs, we conducted two sets of comparative experiments to comprehensively evaluate the efficiency of our proposed method. In the first set of experiments, we kept the input length fixed at 168 and varied the output length to 168/336/720/1440 to study the impact of forecasting length on the actual runtime efficiency of the model. In the second set of experiments, we fixed the output length at 1440 while varying the input length to 168/336/720/1440 to investigate the influence of historical input series length on the actual runtime of the model. The efficiency analysis considered both time and memory aspects. We selected representative methods from each paradigm based on the experimental results in Table 1 as the comparative methods. We fixed the batch size at 32, the model dimension size at 128, and conducted the tests using a single-layer model. The results are shown in Figure 4. Due to the much higher time and memory overhead of TimesNet compared to the other comparative models, we have omitted it from Figure 4 to provide a clearer illustration of the overhead details of the other models. Similarly, FiLM is not included in the time comparison chart.

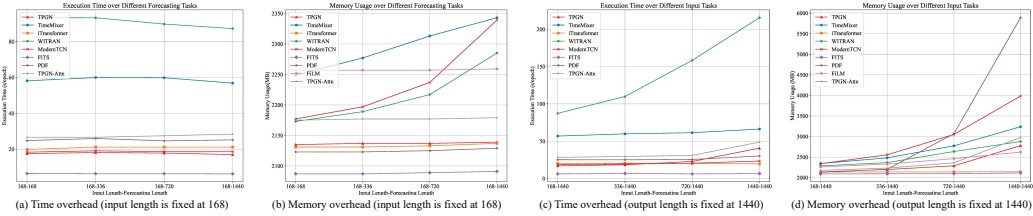

Figure 4: Time and memory overhead of different models.

From Figure 4, it can be observed that although TPGN does not have the lowest time and memory overhead, it achieves a decent level of efficiency in both time and space aspects. It is important to note that TPGN is a model with only one layer, while most of other models require the introduction of deeper layers, which inevitably leads to higher overhead. This undoubtedly further demonstrates that our method not only achieves SOTA performance but also delivers satisfactory efficiency.

## 5 Conclusions

In this paper, we propose a novel general paradigm called Parallel Gated Network (PGN). With its $\mathcal{O}(1)$ information propagation paths and parallel computing capability, PGN achieves faster runtime speed while maintaining the same theoretical complexity as RNN ($\mathcal{O}(L)$). To enhance the application of PGN in long-range time series forecasting tasks, we introduce a novel temporal modeling framework called Temporal PGN (TPGN) with an excellent complexity of $\mathcal{O}(\sqrt{L})$. By employing two branches to separate the modeling of long-term and short-term information, TPGN effectively capture periodicity and local-global semantic information while preserving their respective characteristics. The experimental results on five benchmark datasets demonstrate that our PGN-based framework, TPGN, achieves SOTA performance and high efficiency. These findings further confirm the effectiveness of PGN as the new successor to RNN in long-range time series forecasting tasks.

## Acknowledgments

This work was supported by the National Natural Science Foundation of China (No. 62372031).

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

# A  Limitation and Future Outlook

It is important to acknowledge that our focus in this work was primarily on temporal modeling, without specifically addressing the modeling of relationships between variables. Nevertheless, we can draw inspiration from other methods specialized in variable modeling. Incorporating an additional component to model variable relationships and integrating it into the TPGN framework is a promising direction for better adaptation to multivariate prediction tasks, which we plan to explore in future research. Additionally, we will continue to investigate the broader application of the PGN paradigm as a replacement for RNN in various time series analysis tasks and other research areas.

# B  More Detailed Discussions of Related Works

**Traditional methods** such as ARIMA [Box and Jenkins, 1968], Prophet [Taylor and Letham, 2018], and Holt-Winters [Athanasopoulos and Hyndman, 2020] are often constrained by theoretical assumptions, which limits their applicability in time series forecasting tasks with dynamic data changes. In recent years, **deep neural networks (DNNs)** have made significant advancements in the field of time series analysis. DNNs can be categorized into four main paradigms: **RNN-based**, **CNN-based**, **MLP-based**, and **Transformer-based** methods.

**RNN-based** methods [Hochreiter and Schmidhuber, 1997, Chung et al., 2014, Rangapuram et al., 2018, Salinas et al., 2020] rely on recurrent structures to capture sequential information, which leads to long information propagation paths and brings about various limitations, as discussed in Section 1. In terms of performance, RNN-based methods struggle to capture long-term dependencies effectively. Moreover, their theoretical complexity scales linearly with the sequence length $L$, but their practical efficiency is often low due to sequential computation. Additionally, during training, RNNs are prone to the issues of gradient exploding/vanishing [Pascanu et al., 2013].

To alleviate these issues, some methods have modified the conventional information propagation approach. DilatedRNN [Chang et al., 2017] introduces a multi-scale dilated mechanism, which aggregates information at each time step. Although it can shorten the information propagation path by selecting the branch with the maximum skipping step, the path remains linearly related to the sequence length $L$, which is still relatively long. SlicedRNN [Yu and Liu, 2018] addresses the efficiency problem of RNNs by dividing the sequence into multiple slices for parallel computation. However, the length of the information propagation path remains the same as the traditional RNNs. WITRAN [Jia et al., 2023], as an emerging time series forecasting method, reshapes the sequence into a 2D dimension and performs simultaneous information propagation in both directions. This approach improves computational efficiency and reduces the information propagation path to $\mathcal{O}(\sqrt{L})$. However, it is still relatively long for effective information extraction.

Overall, the limitations imposed by the recurrent structures of RNNs have hindered their further development.

**CNN-based** methods [Bai et al., 2018, Franceschi et al., 2019, Sen et al., 2019] have a theoretical complexity of $\mathcal{O}(L)$, due to their parallel ability, they often exhibit higher practical efficiency compared to RNNs. However, CNNs are typically constrained by limited receptive fields, requiring the stacking of multiple module layers to capture global information from inputs. The number of modular layers grows superlinearly with the sequence length, leading to an information propagation path of $\mathcal{O}(G)$ in CNN-based methods. Here, $G$ is superlinearly related to the sequence length $L$. In the case of the 2D modeling method TimesNet [Wu et al., 2023], where the input lengths in both directions are $\mathcal{O}(\sqrt{L})$, the information propagation path would be $\mathcal{O}(\sqrt{G})$. MICN [Wang et al., 2023] and ModernTCN [Luo and Wang, 2024] effectively reduce the information propagation path by enlarging the receptive field of the convolutional kernel. However, due to their 1D modeling approach, they may not perform as well as TimesNet [Wu et al., 2023] in capturing periodic characteristics.

**MLP-based** methods are highly favored due to their simple structure, resulting in lower computational complexity and information propagation path. This simplicity makes MLP models easy to implement and train, contributing to their popularity. Dlinear and NLinear [Zeng et al., 2023] optimize the original Linear model through the methods of sequence decomposition and re-normalization methods, enabling direct future prediction based on historical inputs. However, due to their limited ability to extract deep semantic information, they may not achieve excellent forecasting performance. TimeMixer [Wang et al., 2024] employs two specialized modules to analyze and predict time series

data from multiple scales. While this approach can effectively capture periodicity, incorporating multiple scales in computations inevitably leads to increased computational costs and training difficulties. FITS [Xu et al., 2024] treats time series prediction as interpolation and transforms the time series into the frequency domain. It operates on the frequency domain using a specially designed block LPF (Low-Pass Filter) and a complex-valued linear layer for final forecasting. However, FITS may still overlook explicit local variations present in the sequence.

**Transformer-based** methods still dominate the majority of the field. The advantage of methods based on point-wise attention mechanism, such as Vanilla-Transformer [Vaswani et al., 2017], Informer [Zhou et al., 2021], and FEDformer [Zhou et al., 2022a], lies in their $\mathcal{O}(1)$ information propagation path. However, previous studies [Wu et al., 2023, Jia et al., 2023] have clearly pointed out their limitation in capturing semantic information from time steps. On the other hand, other methods that utilize non-point-wise attention mechanisms still have other limitations. Although Autoformer [Wu et al., 2021] can capture the periodicity of time series to some extent through sequence decomposition, it is far less direct compared to methods like TimesNet [Wu et al., 2023]. Additionally, its complexity remains high at $\mathcal{O}(L \log L)$. Pyraformer [Liu et al., 2022a], through the special design of its pyramidal structure, is also able to effectively capture the periodic characteristics of sequences. However, it is still constrained by the limitations of the convolution kernel when initializing the node of pyramidal structure. Additionally, Pyraformer still maintains a high complexity of $\mathcal{O}(L)$. PatchTST [Nie et al., 2023] captures local semantic information through patches, where $S$ represents the stride length, and further reduces the complexity to $\mathcal{O}((L/S)^2)$. However, it still cannot directly capture the periodic characteristics of series. iTransformer [Liu et al., 2024] primarily focuses on modeling the relationships between variables, including the relationships between time series variables and external time features. For the temporal dimension, iTransformer adopts a direct patch-based approach, which makes it challenging to effectively extract periodic patterns and other local characteristics. PDF [Dai et al., 2024] also follows the approach of transposing the original 1D sequence into a 2D representation for modeling. Specifically, it utilizes CNNs to process short-term information, which is undoubtedly constrained by the limitations of convolution itself. When it comes to handling long-term periodic information, PDF also adopts a patch-based approach, which may not fully capture all the periodic characteristics present in the sequence.

To highlight the advantages of our proposed PGN paradigm and TPGN framework compared to previous methods, we have organized an information propagation diagram as shown in Figure 1. Based on the above analysis, we further compiled the various strengths, information propagation paths, and theoretical complexities of different models, which are presented in Table 3.

Table 3: Comparison of strengths, complexities and the maximum information propagation paths of different models. $G$ is superlinearly related to the sequence length $L$ and $S$ represents the stride.

| Methods | Capturing non-point-wise semantic information | Directly capturing periodic semantic information | Maximum temporal information propagation path | Complexity of encoder per model layer | Parallel Computing Capability in the in Termporal Dimension |
|---|---|---|---|---|---|
| RNN | ✓ | ✗ | $\mathcal{O}(L)$ | $\mathcal{O}(L)$ | ✗ |
| WITRAN | ✓ | ✓(2D) | $\mathcal{O}(\sqrt{L})$ | $\mathcal{O}(\sqrt{L})$ | ✓̷ |
| CNN | ✓ | ✗ | $\mathcal{O}(G)$ | $\mathcal{O}(L)$ | ✓ |
| MICN | ✓ | ✓̷ (Decomposition) | $\mathcal{O}(1)$ | $\mathcal{O}(L)$ | ✓ |
| TimesNet | ✓ | ✓(2D) | $\mathcal{O}(\sqrt{G})$ | $\mathcal{O}(L)$ | ✓ |
| ModernTCN | ✓ | ✗ | $\mathcal{O}(G)$ | $\mathcal{O}(L/S)$ | ✓ |
| Transformer | ✗ | ✗ | $\mathcal{O}(1)$ | $\mathcal{O}(L^2)$ | ✓ |
| Informer | ✗ | ✗ | $\mathcal{O}(1)$ | $\mathcal{O}(L \log L)$ | ✓ |
| Autoformer | ✓ | ✓̷ (Decomposition) | $\mathcal{O}(1)$ | $\mathcal{O}(L \log L)$ | ✓ |
| Pyraformer | ✓ | ✓(Pyramidal Structure) | $\mathcal{O}(1)$ | $\mathcal{O}(L)$ | ✓ |
| PatchTST | ✓ | ✗ | $\mathcal{O}(1)$ | $\mathcal{O}((L/S)^2)$ | ✓ |
| iTransformer | ✓ | ✗ | $\mathcal{O}(1)$ | $\mathcal{O}(L)$ | ✓ |
| PDF | ✓ | ✓(2D) | $\mathcal{O}(\sqrt{G})$ | $\mathcal{O}(L/S)$ | ✓ |
| PGN (ours) | ✓ | ✗ | $\mathcal{O}(1)$ | $\mathcal{O}(L)$ | ✓ |
| TPGN (ours) | ✓ | ✓(2D) | $\mathcal{O}(1)$ | $\mathcal{O}(\sqrt{L})$ | ✓ |

## C  More Detailed Description of the Datasets

In this section, we will provide a comprehensive overview of the datasets utilized in this paper. (1) *Electricity*[2] (*ECL*) contains the hourly electricity consumption of 321 customers from 2012 to 2014. (2) *Traffic*[3] contains the hourly data the road occupancy rates measured by different sensors on San Francisco Bay area freeways, collected from California Department of Transportation. (3) *ETT*[4] contains the load and oil temperature data recorded every 15 minutes from electricity transformers in two different areas, spans from July 2016 to July 2018. (4) *Weather*[5] contains 21 meteorological indicators (such as air temperature, humidity, etc.) and was recorded every 10 minutes for 2020 whole year.

Due to the varying granularity of data acquisition for each dataset, in order to ensure that they contain the same semantic information for the same task, we followed [Jia et al., 2023] to aggregate them at an hourly level for experimentation. The target value for ECL is 'MT_320', for Traffic is 'Node_862', for ETT is 'oil temperature (OT)', and for Weather is 'wet_bulb'. They are all split into the training set, validation set and test set by the ratio of 6:2:2 during modeling.

Table 4: MSE and MAE with error bars are measured for all methods in long-range forecasting tasks. Each experiment is repeated 5 times.

| Datasets | | ECL | | Traffic | | ETTh1 | | ETTh2 | | Weather | |
|---|---|---|---|---|---|---|---|---|---|---|---|
| Metric | | MSE | MAE | MSE | MAE | MSE | MAE | MSE | MAE | MSE | MAE |
| **TPGN (ours)** | 168-168 | 0.2107±0.00809 | 0.3264±0.00583 | 0.1196±0.00149 | 0.1857±0.00172 | 0.1061±0.00091 | 0.2533±0.00106 | 0.2174±0.01190 | 0.3623±0.00876 | 0.1877±0.00320 | 0.3166±0.00373 |
| | 168-336 | 0.2276±0.00745 | 0.3446±0.00684 | 0.1156±0.00102 | 0.1868±0.00111 | 0.1110±0.00226 | 0.2625±0.00312 | 0.2237±0.00381 | 0.3769±0.00327 | 0.1978±0.00279 | 0.3278±0.00353 |
| | 168-720 | 0.2303±0.01145 | 0.3550±0.01142 | 0.1293±0.00199 | 0.2057±0.00219 | 0.1346±0.00490 | 0.2908±0.00509 | 0.2356±0.00599 | 0.3898±0.00378 | 0.1925±0.00214 | 0.3255±0.00319 |
| | 168-1440 | 0.2484±0.01098 | 0.3775±0.01012 | 0.1390±0.00100 | 0.2114±0.00144 | 0.1343±0.00096 | 0.2941±0.00140 | 0.2514±0.00339 | 0.4070±0.00437 | 0.1786±0.00327 | 0.3184±0.00209 |
| WITRAN | 168-168 | 0.2397±0.00859 | 0.3519±0.00601 | 0.1377±0.00231 | 0.2051±0.00300 | 0.1105±0.00082 | 0.2589±0.00128 | 0.2389±0.00615 | 0.3813±0.00566 | 0.2050±0.00428 | 0.3338±0.00483 |
| | 168-336 | 0.2607±0.00926 | 0.3721±0.00783 | 0.1321±0.00327 | 0.2059±0.00359 | 0.1189±0.00325 | 0.2714±0.00439 | 0.2277±0.00805 | 0.3778±0.00772 | 0.2197±0.00629 | 0.3470±0.00328 |
| | 168-720 | 0.2906±0.00921 | 0.3965±0.00508 | 0.1439±0.00271 | 0.2226±0.00266 | 0.1566±0.00419 | 0.2905±0.00455 | 0.2718±0.02066 | 0.4146±0.01915 | 0.2538±0.00436 | 0.3796±0.00405 |
| | 168-1440 | 0.3255±0.02833 | 0.4302±0.02040 | 0.1611±0.00713 | 0.2369±0.00601 | 0.1541±0.01205 | 0.3157±0.01239 | 0.3350±0.03795 | 0.4624±0.02629 | 0.2695±0.00660 | 0.3966±0.00417 |
| ModernTCN | 168-168 | 0.2473±0.01112 | 0.3437±0.00856 | 0.1473±0.00124 | 0.2212±0.00154 | 0.1210±0.00612 | 0.2694±0.00613 | 0.2564±0.00293 | 0.3980±0.00265 | 0.2692±0.00831 | 0.4088±0.00653 |
| | 168-336 | 0.3110±0.00451 | 0.3887±0.00474 | 0.1410±0.00165 | 0.2214±0.00246 | 0.1342±0.00262 | 0.2884±0.00343 | 0.2918±0.01222 | 0.4312±0.00844 | 0.2916±0.01068 | 0.4314±0.00844 |
| | 168-720 | 0.3624±0.01165 | 0.4478±0.00711 | 0.1574±0.00220 | 0.2389±0.00230 | 0.1676±0.00653 | 0.3238±0.00738 | 0.3991±0.01156 | 0.5022±0.00678 | 0.4279±0.00790 | 0.4812±0.00333 |
| | 168-1440 | 0.5307±0.01639 | 0.5573±0.00637 | 0.1980±0.00169 | 0.2739±0.00265 | 0.2756±0.02847 | 0.4247±0.02444 | 0.8537±0.09298 | 0.7437±0.04061 | 0.7049±0.02628 | 0.6351±0.00672 |
| TimesNet | 168-168 | 0.2825±0.00960 | 0.3797±0.00798 | 0.1490±0.00412 | 0.2293±0.00686 | 0.1133±0.00253 | 0.2612±0.00377 | 0.2655±0.01248 | 0.4051±0.01202 | 0.2420±0.00614 | 0.3608±0.00466 |
| | 168-336 | 0.3505±0.00624 | 0.4253±0.00428 | 0.1499±0.00136 | 0.2356±0.00298 | 0.1202±0.00237 | 0.2732±0.00359 | 0.2725±0.00610 | 0.4163±0.00564 | 0.2821±0.01949 | 0.3885±0.01692 |
| | 168-720 | 0.4261±0.02519 | 0.4686±0.01447 | 0.1621±0.00715 | 0.2471±0.00740 | 0.1458±0.00172 | 0.3059±0.00189 | 0.3186±0.03843 | 0.4465±0.02545 | 0.2941±0.00468 | 0.4013±0.00360 |
| | 168-1440 | 0.6688±0.11196 | 0.6102±0.05236 | 0.1691±0.01544 | 0.2517±0.01590 | 0.1543±0.00484 | 0.3119±0.00482 | 0.3839±0.05262 | 0.4933±0.03400 | 0.2988±0.01396 | 0.4092±0.00742 |
| MICN | 168-168 | 0.3168±0.01684 | 0.3797±0.01116 | 0.2418±0.00578 | 0.3537±0.00560 | 0.1257±0.01332 | 0.2803±0.01514 | 0.2734±0.01795 | 0.4162±0.01662 | 0.2231±0.00219 | 0.3489±0.00404 |
| | 168-336 | 0.3002±0.00979 | 0.4253±0.00853 | 0.2420±0.00709 | 0.3568±0.00685 | 0.1422±0.02929 | 0.3006±0.03410 | 0.3017±0.04440 | 0.4429±0.03426 | 0.2663±0.00330 | 0.3837±0.00602 |
| | 168-720 | 0.4453±0.03694 | 0.4686±0.02169 | 0.2488±0.00996 | 0.3592±0.01086 | 0.1609±0.00770 | 0.3200±0.00866 | 0.4770±0.04330 | 0.5602±0.02730 | 0.3077±0.00374 | 0.4202±0.00720 |
| | 168-1440 | 0.8784±0.23697 | 0.6102±0.10350 | 0.2817±0.01048 | 0.3818±0.00895 | 0.1444±0.01543 | 0.3032±0.01590 | 0.4876±0.02074 | 0.5602±0.01464 | 0.4306±0.02393 | 0.4994±0.01621 |
| FITS | 168-168 | 0.2598±0.00013 | 0.3573±0.00012 | 0.1498±0.00006 | 0.2134±0.00013 | 0.1089±0.00013 | 0.2556±0.00017 | 0.2547±0.00029 | 0.3947±0.00033 | 0.2423±0.00021 | 0.3561±0.00033 |
| | 168-336 | 0.3072±0.00080 | 0.3938±0.00082 | 0.1445±0.00007 | 0.2148±0.00009 | 0.1162±0.00005 | 0.2682±0.00004 | 0.2874±0.00070 | 0.4250±0.00064 | 0.2977±0.00033 | 0.4007±0.00038 |
| | 168-720 | 0.3504±0.00030 | 0.4366±0.00027 | 0.1603±0.00018 | 0.2330±0.00025 | 0.1544±0.00007 | 0.3109±0.00007 | 0.3983±0.00034 | 0.5023±0.00029 | 0.4044±0.00026 | 0.4758±0.00036 |
| | 168-1440 | 0.5176±0.00089 | 0.5591±0.00058 | 0.1845±0.00020 | 0.2571±0.00014 | 0.2319±0.00007 | 0.3863±0.00006 | 0.7852±0.00109 | 0.7256±0.00062 | 0.7027±0.00018 | 0.6555±0.00027 |
| TimeMixer | 168-168 | 0.2804±0.00607 | 0.3792±0.00257 | 0.1340±0.00363 | 0.2124±0.00380 | 0.1110±0.00125 | 0.2587±0.00181 | 0.2507±0.00774 | 0.3936±0.00637 | 0.2275±0.00259 | 0.3466±0.00248 |
| | 168-336 | 0.3183±0.01196 | 0.4029±0.00794 | 0.1298±0.00147 | 0.2147±0.00238 | 0.1209±0.01122 | 0.2716±0.00937 | 0.2642±0.00216 | 0.4085±0.00216 | 0.2775±0.00230 | 0.3836±0.00239 |
| | 168-720 | 0.3835±0.01693 | 0.4560±0.01092 | 0.1396±0.00265 | 0.2285±0.00383 | 0.1362±0.00266 | 0.2927±0.00232 | 0.3259±0.02873 | 0.4510±0.01920 | 0.2873±0.00559 | 0.3931±0.00428 |
| | 168-1440 | 0.6857±0.07838 | 0.6194±0.03071 | 0.1547±0.00199 | 0.2392±0.00411 | 0.1480±0.00591 | 0.3068±0.00645 | 0.3937±0.01583 | 0.5040±0.01159 | 0.3010±0.02115 | 0.4078±0.01101 |
| DLinear | 168-168 | 0.2606±0.00138 | 0.3579±0.00117 | 0.1519±0.00017 | 0.2195±0.00023 | 0.1122±0.00081 | 0.2605±0.00106 | 0.2556±0.00080 | 0.3944±0.00091 | 0.2421±0.00302 | 0.3578±0.00240 |
| | 168-336 | 0.3080±0.00167 | 0.3946±0.00149 | 0.1468±0.00012 | 0.2210±0.00017 | 0.1251±0.00086 | 0.2794±0.00101 | 0.2891±0.00143 | 0.4256±0.00082 | 0.2918±0.00149 | 0.3975±0.00112 |
| | 168-720 | 0.3515±0.00175 | 0.4374±0.00130 | 0.1629±0.00043 | 0.2389±0.00045 | 0.1919±0.00091 | 0.3465±0.00094 | 0.4090±0.00341 | 0.5090±0.00206 | 0.3915±0.00389 | 0.4739±0.00195 |
| | 168-1440 | 0.5300±0.00062 | 0.5681±0.00044 | 0.1890±0.00145 | 0.2640±0.00120 | 0.3606±0.00179 | 0.4939±0.00136 | 0.7921±0.00200 | 0.7262±0.00193 | 0.5837±0.00267 | 0.6177±0.00134 |
| iTransformer | 168-168 | 0.2479±0.00185 | 0.3516±0.00149 | 0.1343±0.00025 | 0.2083±0.00054 | 0.1112±0.00116 | 0.2598±0.00133 | 0.2630±0.00436 | 0.4053±0.00398 | 0.2636±0.00410 | 0.4056±0.00360 |
| | 168-336 | 0.3128±0.00332 | 0.3974±0.00285 | 0.1366±0.00060 | 0.2221±0.00038 | 0.1203±0.00172 | 0.2709±0.00093 | 0.2658±0.00583 | 0.4092±0.00423 | 0.2658±0.00583 | 0.4092±0.00423 |
| | 168-720 | 0.3660±0.00690 | 0.4438±0.00395 | 0.1402±0.00134 | 0.2265±0.00188 | 0.1423±0.00271 | 0.3020±0.00293 | 0.2951±0.00618 | 0.4312±0.00435 | 0.2754±0.00312 | 0.3900±0.00371 |
| | 168-1440 | 0.7028±0.06470 | 0.6348±0.03342 | 0.1519±0.00054 | 0.2321±0.00135 | 0.1520±0.00875 | 0.3107±0.00860 | 0.3806±0.00889 | 0.4894±0.00504 | 0.3007±0.01103 | 0.4113±0.00881 |
| PDF | 168-168 | 0.2483±0.00301 | 0.3491±0.00269 | 0.1397±0.00076 | 0.2119±0.00138 | 0.1115±0.00091 | 0.2579±0.00083 | 0.2606±0.00287 | 0.4012±0.00282 | 0.2557±0.00239 | 0.3989±0.00238 |
| | 168-336 | 0.3094±0.00244 | 0.3902±0.00311 | 0.1351±0.00090 | 0.2132±0.00078 | 0.1207±0.00298 | 0.2725±0.00328 | 0.3064±0.01527 | 0.4440±0.01142 | 0.3065±0.01527 | 0.4440±0.01142 |
| | 168-720 | 0.3541±0.00379 | 0.4423±0.00267 | 0.1502±0.00058 | 0.2290±0.00063 | 0.1720±0.00424 | 0.3278±0.00418 | 0.4546±0.03786 | 0.5415±0.02478 | 0.3988±0.00271 | 0.4673±0.00195 |
| | 168-1440 | 0.9029±0.06926 | 0.6913±0.02764 | 0.2074±0.00210 | 0.2779±0.00139 | 0.2792±0.00995 | 0.4272±0.00842 | 0.8699±0.05495 | 0.7651±0.02818 | 0.7334±0.03396 | 0.6557±0.01771 |
| Basisformer | 168-168 | 0.3116±0.00515 | 0.4026±0.00460 | 0.1634±0.00380 | 0.2553±0.00749 | 0.1169±0.00338 | 0.2646±0.00389 | 0.2806±0.00869 | 0.4138±0.00748 | 0.2301±0.00378 | 0.3541±0.00413 |
| | 168-336 | 0.4844±0.03133 | 0.4824±0.00985 | 0.1544±0.00177 | 0.2493±0.00317 | 0.1227±0.00393 | 0.2734±0.00401 | 0.2697±0.00782 | 0.4132±0.00701 | 0.2593±0.00656 | 0.3751±0.00873 |
| | 168-720 | 0.6448±0.13800 | 0.5653±0.05703 | 0.1538±0.00771 | 0.2490±0.00973 | 0.1521±0.00289 | 0.3121±0.00305 | 0.3034±0.01322 | 0.4372±0.00843 | 0.2834±0.00726 | 0.3916±0.00630 |
| | 168-1440 | 0.6368±0.07819 | 0.5967±0.04075 | 0.1735±0.00437 | 0.2654±0.00416 | 0.1664±0.01616 | 0.3230±0.01100 | 0.3963±0.02934 | 0.4996±0.01896 | 0.2959±0.01179 | 0.4095±0.00907 |
| PatchTST | 168-168 | 0.2980±0.00339 | 0.3832±0.00341 | 0.1622±0.01189 | 0.2320±0.00878 | 0.1212±0.00480 | 0.2704±0.00375 | 0.2582±0.00722 | 0.3983±0.00598 | 0.2469±0.00797 | 0.3597±0.00569 |
| | 168-336 | 0.3446±0.01157 | 0.4094±0.00653 | 0.1641±0.01070 | 0.2364±0.00646 | 0.1287±0.00483 | 0.2808±0.00483 | 0.3206±0.01683 | 0.4514±0.01199 | 0.3040±0.00457 | 0.4049±0.00439 |
| | 168-720 | 0.4324±0.01333 | 0.4782±0.00525 | 0.1770±0.01043 | 0.2548±0.00976 | 0.1727±0.00601 | 0.3297±0.00505 | 0.4398±0.03582 | 0.5304±0.02195 | 0.4023±0.01475 | 0.4662±0.00877 |
| | 168-1440 | 0.7349±0.08816 | 0.6464±0.03219 | 0.2139±0.00967 | 0.2875±0.00690 | 0.3206±0.02700 | 0.4561±0.02092 | 0.8339±0.02378 | 0.7330±0.01073 | 0.7150±0.04967 | 0.6336±0.02111 |
| FiLM | 168-168 | 0.2587±0.00032 | 0.3557±0.00029 | 0.1501±0.00067 | 0.2143±0.00139 | 0.1091±0.00007 | 0.2558±0.00013 | 0.2546±0.00141 | 0.3942±0.00161 | 0.2426±0.00037 | 0.3544±0.00036 |
| | 168-336 | 0.3062±0.00030 | 0.3922±0.00014 | 0.1453±0.00033 | 0.2165±0.00071 | 0.1187±0.00047 | 0.2708±0.00048 | 0.2894±0.00718 | 0.4263±0.00566 | 0.2981±0.00036 | 0.3988±0.00033 |
| | 168-720 | 0.3486±0.00088 | 0.4349±0.00074 | 0.1617±0.00038 | 0.2358±0.00052 | 0.1717±0.00106 | 0.3266±0.00119 | 0.4039±0.00555 | 0.5061±0.00400 | 0.4093±0.00102 | 0.4737±0.00092 |
| | 168-1440 | 0.5146±0.00127 | 0.5565±0.00065 | 0.1861±0.00157 | 0.2615±0.00148 | 0.3056±0.00208 | 0.4494±0.00183 | 0.7843±0.02010 | 0.7210±0.01035 | 0.7028±0.00121 | 0.6453±0.00114 |

## D  Experimental Setup Details

To ensure a fair comparison of the performance of each model, we set the same search space for the common parameters included in each model. Specifically, (1) The model dimension size $d_m$ is set to: 2, 4, 8, 16, 32, 64, 128, 256, 512, 1024. (2) The number of layers for the model's encoder and decoder is set to: 1, 2, 3. (3) The number of heads for the attention mechanism is set to: 1, 2, 4, 8. In addition, for the individual hyperparameters specific to each model, we also referred to their

---

[2] https://archive.ics.uci.edu/ml/datasets/ElectricityLoadDiagrams20112014
[3] http://pems.dot.ca.gov
[4] https://github.com/zhouhaoyi/ETDataset
[5] https://www.bgc-jena.mpg.de/wetter/

respective original papers and conducted parameter search accordingly. This ensured that we took into account the optimal configurations for each model in our experiments. For models that have variants, such as MICN-regre and MICN-mean, we treat the variants as separate hyperparameters and include them in the search process. The aforementioned procedures ensure that the reported results represent the optimal performance of each model under the same comparison conditions.

TPGN and all baselines were trained using L2 loss and the ADAM optimizer [Kingma and Ba, 2014] with an initial learning rate of $10^{-3}$. All of them are implemented using PyTorch [Paszke et al., 2019] and conducted on NVIDIA RTX A4000 16GB GPUs. The batch size was set to 32 and the maximum training epochs was set to 25. If there was no degradation in loss on the validation set after 5 epochs, the training process would be early stopped. We saved the model with the lowest loss on the validation set for final testing. The mean square error (MSE) and mean absolute error (MAE) are used as metrics. We followed [Jia et al., 2023] and set the seed to 2023 to ensure the reproducibility of the results. All experiments are repeated 5 times and we set the mean of the metrics as the final results, as shown in Table 1.

# E  Error Bars

During the model training process, we conducted tests using the parameters that achieved the lowest loss on the validation set. This process was repeated five times, and the error bars were calculated and presented in Table 4. The results in Table 4 clearly demonstrate the stability of our proposed method, TPGN, further confirming its superior overall performance compared to other baseline models, being SOTA approach.

# F  Comprehensive Experiment for Performance Variations with Different Forecasting Lengths

We conducted a comprehensive experiment to analyze the performance variation of the models across different forecasting lengths, as mentioned in Subsection 4.1. According to the experimental results in Table 1, we selected representative methods from different paradigms: WITRAN (RNN-Based), TimesNet (CNN-Based), TimeMixer (MLP-Based), and iTransformer (Transformer-Based), and compared them with TPGN. The results of dataset ECL have already been presented in Figure 3. Therefore, in this section, we present the other dataset results, as shown in Figure 5-Figure 8, and further analysis them.

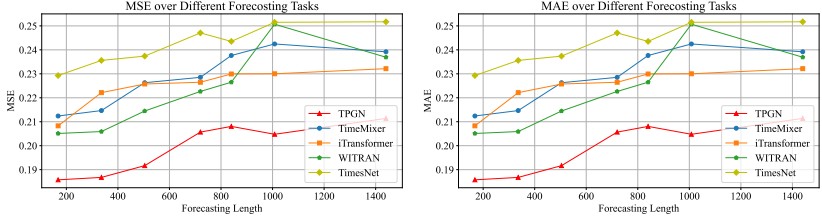

Figure 5: Experimental results with different forecasting lengths on the Traffic dataset.

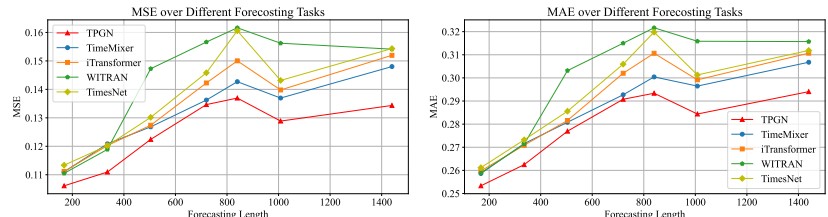

Figure 6: Experimental results with different forecasting lengths on the ETTh$_1$ dataset.

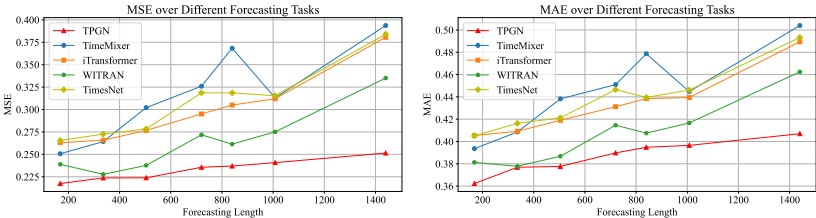

Figure 7: Experimental results with different forecasting lengths on the ETTh$_2$ dataset.

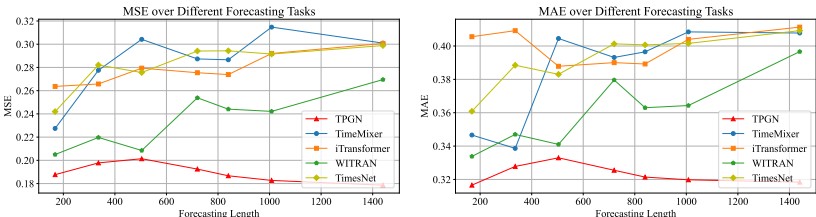

Figure 8: Experimental results with different forecasting lengths on the Weather dataset.

Through the results in Figure 3, Figure 5, Figure 6, Figure 7, and Figure 8, we can clearly observe that as the sequence length increases, all models show varying degrees of performance degradation across different datasets. However, TPGN not only consistently maintains SOTA performance across all tasks but also exhibits slower performance decay with increasing forecasting lengths in most cases. This demonstrates the significant advantage of TPGN in predicting longer-range tasks, further confirming its ability to effectively extract information from limited inputs and apply them well to prediction tasks.

## G   Robustness Analysis

To assess the robustness of TPGN, we conducted experiments following the settings of MICN [Wang et al., 2023] and WITRAN [Jia et al., 2023], and introduced a simple white noise injection. Specifically, a random proportion $\varepsilon$ of data was selected from the original input sequence, and random perturbations within the range $[-2X_i, 2X_i]$ were applied to the selected data, where $X_i$ represents the original data. Subsequently, the injected noisy data was used for training, and we recorded the MSE and MAE metrics in Table 5.

It can be observed that as the perturbation ratio increases, there is a slight increase in the MSE and MAE metrics in terms of forecasting. This indicates that TPGN exhibits good robustness in handling data with low noise levels (up to 10%) and has a significant advantage in effectively handling various abnormal data fluctuations.

## H   Parameter Sensitivity

In our model, we have only two hyperparameters. One is the number of hidden units, denoted as $d_{\mathrm{m}}$, which is a common hyperparameter in many models. Its value is determined through parameter search and further validation on the validation set. During our parameter search process, we have observed that different models exhibit significant variations in the optimal $d_{\mathrm{m}}$ values for the same task. Similarly, even for the same model, this variability persists when facing different tasks. Therefore, in this paper, we focus our analysis solely on the parameter $norm$, as it showcases the notable differences and its impact on the model's performance.

In the Subsection 3.2, we mentioned that the parameter $norm$ should be determined based on the dataset, which has been previously acknowledged in related works [Jia et al., 2023]. However, to validate the appropriateness of our $norm$ selection in the preparatory work, we conducted a statistical analysis on the variations across different task datasets, as shown in Table 6.

Table 5: **Robustness experiments of TPGN**'s forecasting results. Different $\varepsilon$ indicates different proportions of noise injection.

| Tasks | | 168-168 | | 168-336 | | 168-720 | | 168-1440 | |
|---|---|---|---|---|---|---|---|---|---|
| Metric | | MSE | MAE | MSE | MAE | MSE | MAE | MSE | MAE |
| ECL | $\varepsilon = 0\%$ | 0.2107 | 0.3264 | 0.2276 | 0.3446 | 0.2303 | 0.3550 | 0.2484 | 0.3775 |
| | $\varepsilon = 1\%$ | 0.2116 | 0.3248 | 0.2277 | 0.3447 | 0.2311 | 0.3557 | 0.2484 | 0.3776 |
| | $\varepsilon = 5\%$ | 0.2117 | 0.3270 | 0.2295 | 0.3464 | 0.2314 | 0.3553 | 0.2495 | 0.3783 |
| | $\varepsilon = 10\%$ | 0.2142 | 0.3288 | 0.2303 | 0.3466 | 0.2332 | 0.3561 | 0.2497 | 0.3784 |
| Traffic | $\varepsilon = 0\%$ | 0.1196 | 0.1857 | 0.1156 | 0.1868 | 0.1293 | 0.2057 | 0.1390 | 0.2114 |
| | $\varepsilon = 1\%$ | 0.1201 | 0.1861 | 0.1154 | 0.1869 | 0.1295 | 0.2059 | 0.1393 | 0.2120 |
| | $\varepsilon = 5\%$ | 0.1219 | 0.1899 | 0.1176 | 0.1913 | 0.1296 | 0.2064 | 0.1439 | 0.2200 |
| | $\varepsilon = 10\%$ | 0.1261 | 0.1964 | 0.1205 | 0.1960 | 0.1324 | 0.2119 | 0.1472 | 0.2241 |
| ETTh$_1$ | $\varepsilon = 0\%$ | 0.1061 | 0.2533 | 0.1110 | 0.2625 | 0.1346 | 0.2908 | 0.1343 | 0.2941 |
| | $\varepsilon = 1\%$ | 0.1061 | 0.2534 | 0.1110 | 0.2628 | 0.1348 | 0.2908 | 0.1350 | 0.2944 |
| | $\varepsilon = 5\%$ | 0.1067 | 0.2546 | 0.1111 | 0.2631 | 0.1355 | 0.2916 | 0.1407 | 0.2990 |
| | $\varepsilon = 10\%$ | 0.1068 | 0.2549 | 0.1113 | 0.2630 | 0.1365 | 0.2932 | 0.1446 | 0.3023 |
| ETTh$_2$ | $\varepsilon = 0\%$ | 0.2174 | 0.3623 | 0.2237 | 0.3769 | 0.2356 | 0.3898 | 0.2514 | 0.4070 |
| | $\varepsilon = 1\%$ | 0.2178 | 0.3629 | 0.2237 | 0.3770 | 0.2358 | 0.3900 | 0.2521 | 0.4086 |
| | $\varepsilon = 5\%$ | 0.2180 | 0.3632 | 0.2240 | 0.3772 | 0.2358 | 0.3901 | 0.2521 | 0.4090 |
| | $\varepsilon = 10\%$ | 0.2187 | 0.3639 | 0.2242 | 0.3775 | 0.2360 | 0.3904 | 0.2525 | 0.4097 |
| Weather | $\varepsilon = 0\%$ | 0.1877 | 0.3166 | 0.1978 | 0.3278 | 0.1925 | 0.3255 | 0.1786 | 0.3184 |
| | $\varepsilon = 1\%$ | 0.1884 | 0.3184 | 0.1980 | 0.3282 | 0.1944 | 0.3273 | 0.1788 | 0.3186 |
| | $\varepsilon = 5\%$ | 0.1886 | 0.3184 | 0.1987 | 0.3295 | 0.1950 | 0.3278 | 0.1788 | 0.3188 |
| | $\varepsilon = 10\%$ | 0.1889 | 0.3185 | 0.1993 | 0.3320 | 0.1952 | 0.3281 | 0.1797 | 0.3197 |

Through statistical analysis of the dataset, we have found that there are some differences between the training set and the validation set. These differences are due to the inherent characteristics of time series data and are considered normal fluctuations. If the variances of the training set and validation set are roughly in the same range, it can be assumed that their fluctuations are roughly consistent. In this case, there are no significant differences in data distribution between the training set and validation set, so there is no need for normalization. The hyperparameter $norm$ should be set to 0.

However, when there is a relatively large difference in variance between the training set and the validation set (approximately twice or half), it indicates a significant difference in data distribution between the two sets. In such cases, the hyperparameter $norm$ should be set to 1 to facilitate better training of the model.

Additionally, weather datasets are a special case because they contain negative values, which leads to their mean being close to zero and a significant difference between the variance and mean. This is reasonable for weather data. On the other hand, traffic datasets do not have negative values. Therefore, even if the variances of the training set and validation set are similar, their differences can still be observed when analyzed in conjunction with the mean.

Table 6: The distribution (Mean and STD) of dataset in the training and validation sets.

| Datasets | ECL | | | Traffic | | | ETTh1 | | | ETTh2 | | | Weather | | |
|---|---|---|---|---|---|---|---|---|---|---|---|---|---|---|---|
| Tasks | training set | validation set | norm | training set | validation set | norm | training set | validation set | norm | training set | validation set | norm | training set | validation set | norm |
| 168-168 | 3425.7773±562.8469 | 3043.2100±382.1743 | | 0.0288±0.0170 | 0.0341±0.0200 | | 17.3383±8.5791 | 7.6739±4.2814 | | 29.0542±12.1271 | 20.8957±9.1261 | | 0.3893±6.6630 | 1.3814±7.7366 | |
| 168-336 | 3426.0938±563.6643 | 3031.5738±362.8032 | 0 | 0.0287±0.0170 | 0.0340±0.0200 | 1 | 17.3711±8.6347 | 7.8227±4.3176 | 1 | 28.8913±12.1396 | 21.2674±9.1471 | 0 | 0.3712±6.6834 | 1.5999±7.6813 | 0 |
| 168-720 | 3423.0552±563.4796 | 3026.3346±343.3309 | | 0.0287±0.0169 | 0.0339±0.0199 | | 17.4107±8.7921 | 8.5642±4.0161 | | 28.4763±12.1533 | 22.7715±8.6228 | | 0.3372±6.7363 | 2.1966±7.4464 | |
| 168-1440 | 3407.1474±558.3798 | 3017.0031±327.7298 | | 0.0286±0.0169 | 0.0336±0.0197 | | 17.1370±9.0535 | 10.0193±3.5586 | | 27.1838±11.6251 | 26.0480±7.1308 | | 0.3601±6.8383 | 3.3196±6.9377 | |

# I Case Study

To highlight the advantages of TPGN in long-range time series forecasting tasks, we compared TPGN's showcase with the showcases of the second-best and third-best models, selected based on MSE as the evaluation metric, for each dataset. The forecasting results showcases are shown in Figure 9 to Figure 23.

The comparison of the aforementioned cases clearly demonstrates the superiority of our method in terms of forecasting results. This unequivocally reaffirms the SOTA performance of TPGN across various domains. Moreover, it provides further evidence that PGN, as the novel successor to RNNs, can be effectively applied to long-range time series forecasting tasks.

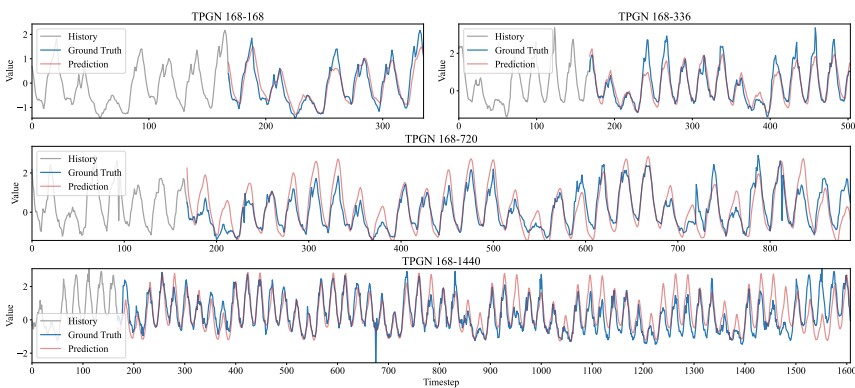

Figure 9: Forecasting cases of TPGN for all tasks in dataset ECL.

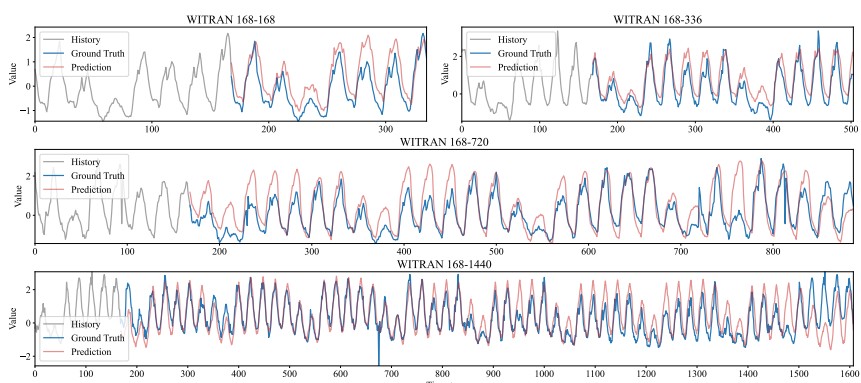

Figure 10: Forecasting cases of WITRAN for all tasks in dataset ECL.

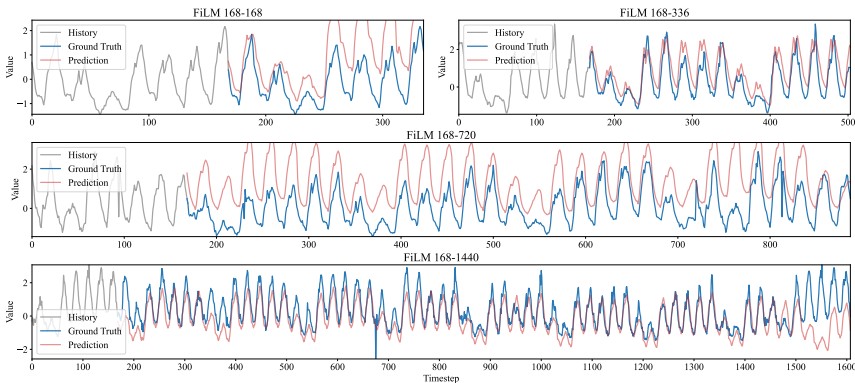

Figure 11: Forecasting cases of FiLM for all tasks in dataset ECL.

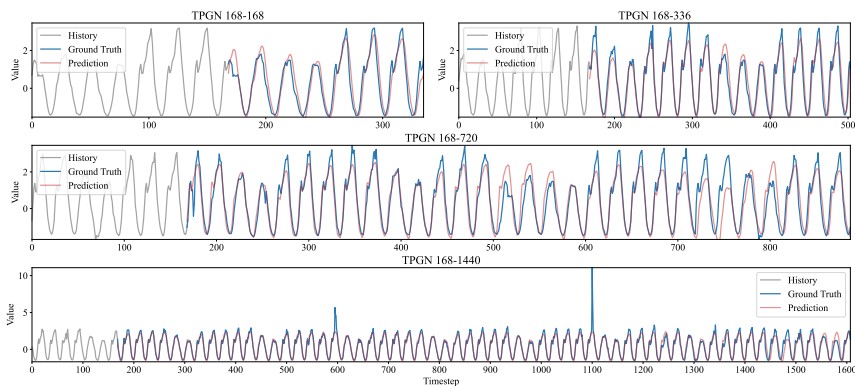

Figure 12: Forecasting cases of TPGN for all tasks in dataset Traffic.

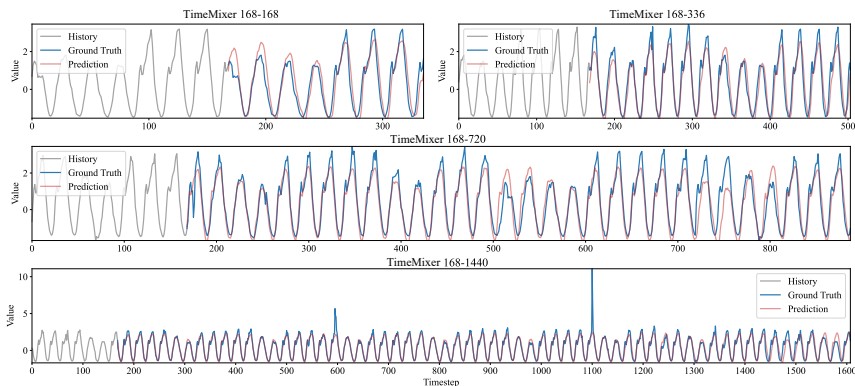

Figure 13: Forecasting cases of TimeMixer for all tasks in dataset Traffic.

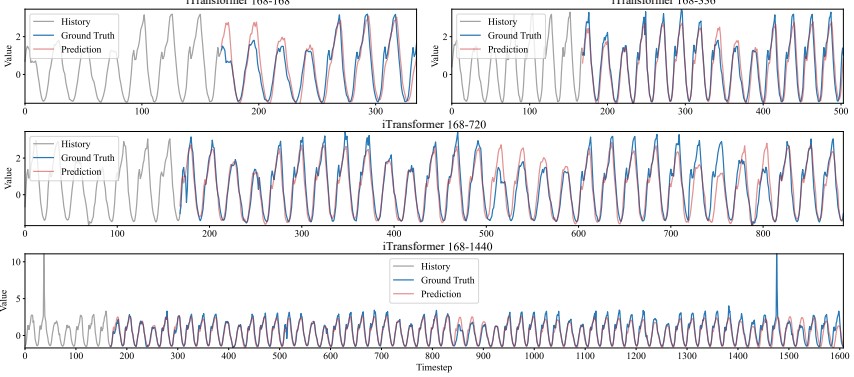

Figure 14: Forecasting cases of iTransformer for all tasks in dataset Traffic.

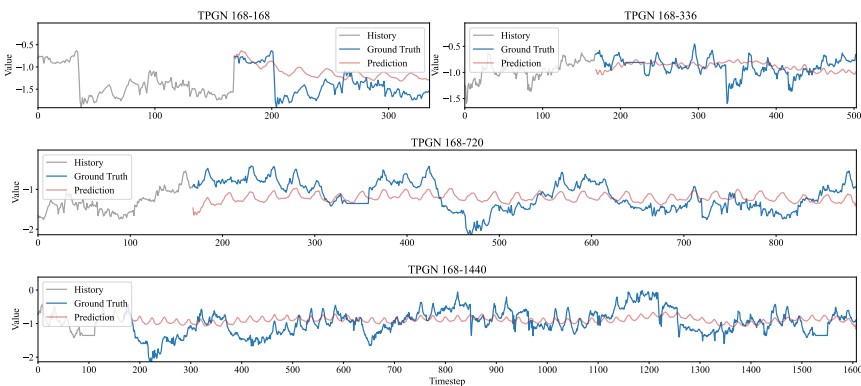

Figure 15: Forecasting cases of TPGN for all tasks in dataset ETTh$_1$.

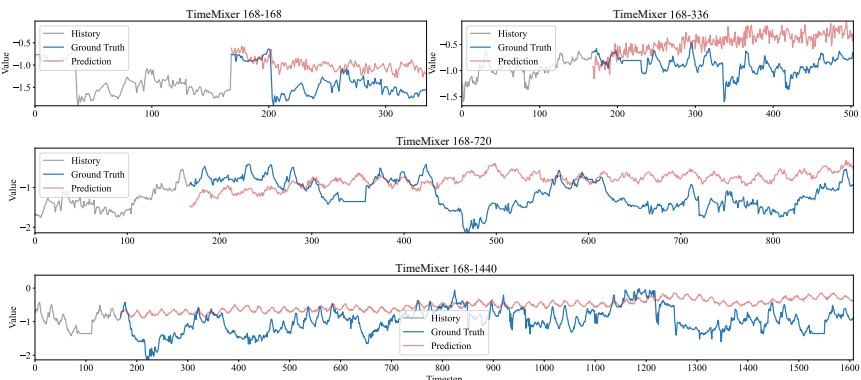

Figure 16: Forecasting cases of TimeMixer for all tasks in dataset ETTh$_1$.

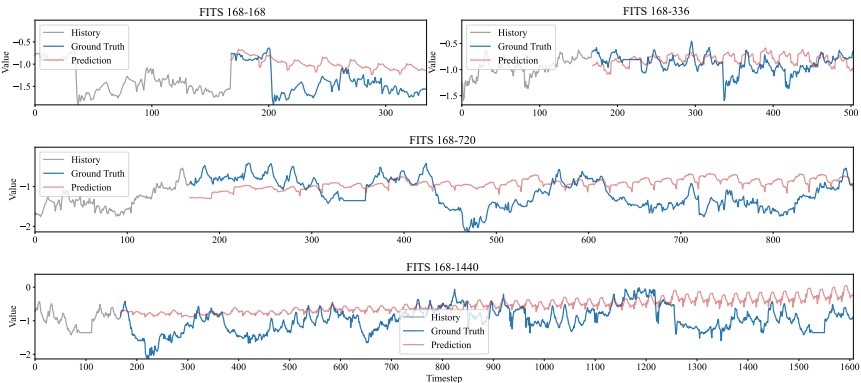

Figure 17: Forecasting cases of FITS for all tasks in dataset ETTh$_1$.

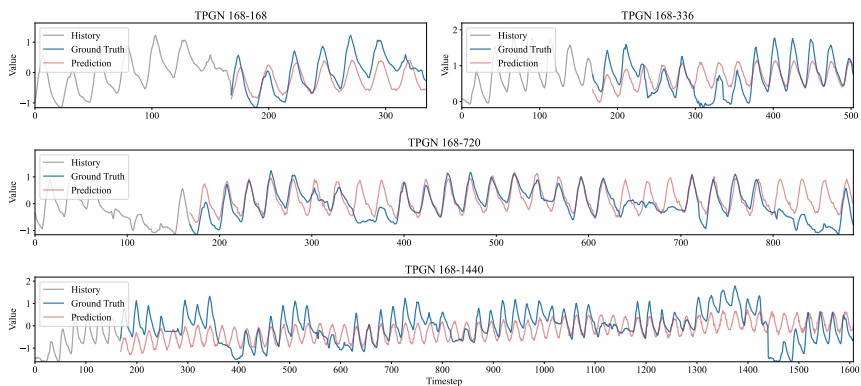

Figure 18: Forecasting cases of TPGN for all tasks in dataset ETTh$_2$.

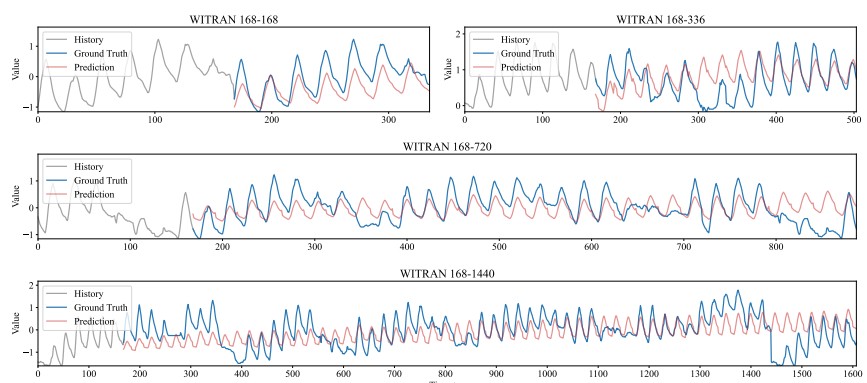

Figure 19: Forecasting cases of WITRAN for all tasks in dataset ETTh$_2$.

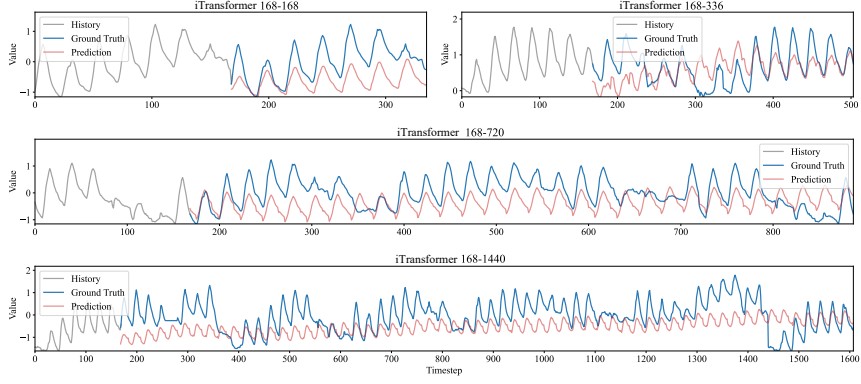

Figure 20: Forecasting cases of iTransformer for all tasks in dataset ETTh$_2$.

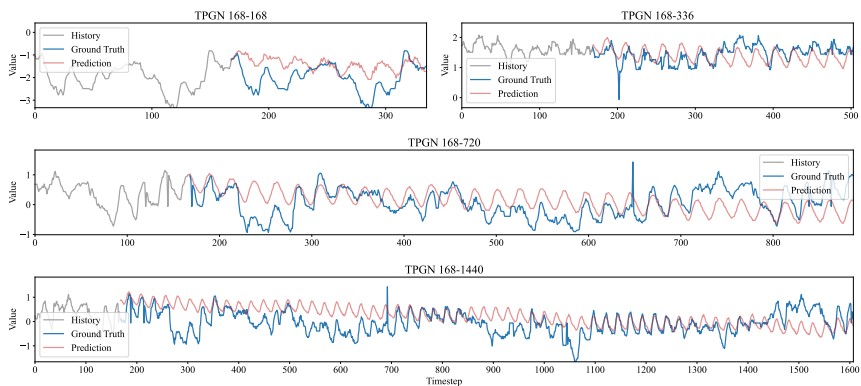

Figure 21: Forecasting cases of TPGN for all tasks in dataset Weather.

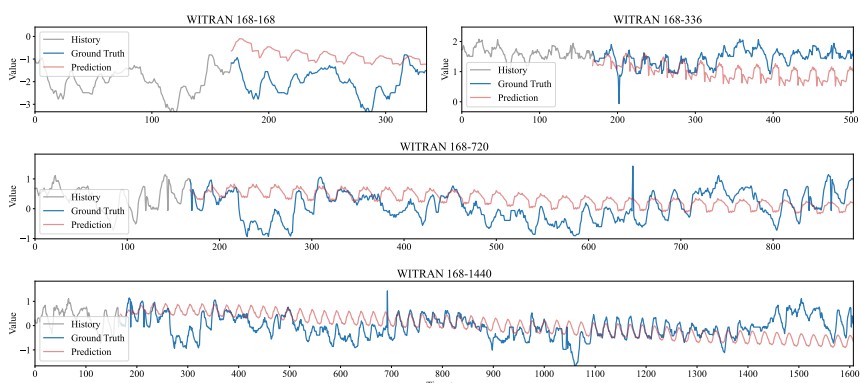

Figure 22: Forecasting cases of WITRAN for all tasks in dataset Weather.

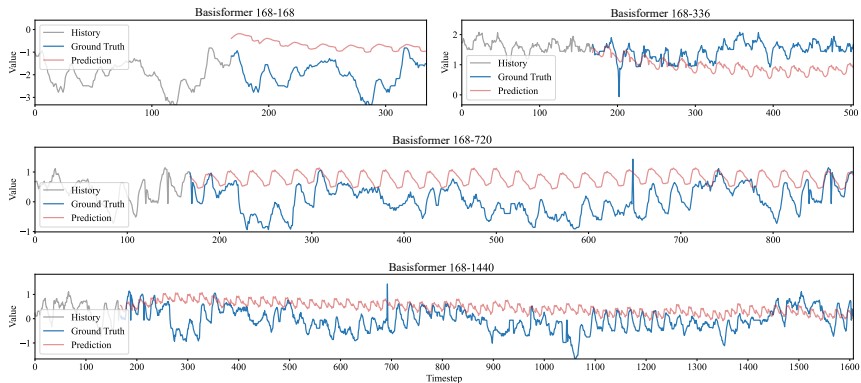

Figure 23: Forecasting cases of Basisformer for all tasks in dataset Weather.

