# OpenReview forum: "PGN: The RNN's New Successor is Effective for Long-Range Time Series Forecasting"
_NeurIPS.cc/2024/Conference — NeurIPS 2024 poster_

### Official Review · Reviewer_Bhsw · 2024-07-03

**Soundness:** 3
**Presentation:** 2
**Contribution:** 2
**Rating:** 6
**Confidence:** 4

**Summary:**

This paper proposed a Parallel Gated Network (PGN) as a successor to RNN, featuring a Historical Information Extraction (HIE) layer to directly capture information from previous time steps. Additionally, it introduces a Temporal PGN (TPGN) framework with two branches to capture both long-term periodic and short-term semantic patterns, demonstrating state-of-the-art performance in long-range time series forecasting.

**Strengths:**

1. This paper compares a variety of cutting-edge methods.
2. The experiments are generally thorough.

**Weaknesses:**

1. The major issue with this paper is the lack of analysis and comparison with significant literature. The entire paper's premise is the traditional RNN's failure in long-term sequence problems due to the long information propagation paths of its recurrent structure. However, as far as I know, SegRNN[1] has already addressed these shortcomings of traditional RNN in long-term forecasting through segmented iteration and parallel prediction. Yet, there is no discussion on this in the paper. Please compare your method with it and clarify your differences and advantages.

2. In Section 2, you should distinguish between Linear-based and MLP-based methods. The former has only single-layer parameter connections, while the latter has multiple layers and can learn non-linear features due to the presence of activation functions. Methods like DLinear and FITS should be classified as Linear-based methods.

3. The description of HIE is unclear: (i) The process shown in Figure 2(a) suggests first performing linear mapping and then zero-padding, which conflicts with Equation 1 in the paper, where H = HIE(Padding(X)), and the actual code. It is recommended to modify Figure 2 to make this clearer. (ii) Line 169 describes that “HIE(·) is a linear layer,” but in practice, the behavior of HIE is more like a sliding aggregation operation of CNN (or TCN) rather than a sorely linear mapping. Given **(ii)**, calling the proposed method RNN-based is debatable since it is more likely TCN-based.

4. You should include an ablation analysis of the normalization layer, explaining its impact on TPGN achieving state-of-the-art results.

5. Although the authors provide source code, it does not include the hyperparameter settings required to reproduce the key results in the paper, meaning there is no directly runnable script. Are the hyperparameters in the main results all defaults? For instance, is TPGN_period=24? If not, providing a complete script file that can be run directly is necessary.


[1] Lin, S., Lin, W., Wu, W., Zhao, F., Mo, R., & Zhang, H. (2023). Segrnn: Segment recurrent neural network for long-term time series forecasting. arXiv preprint arXiv:2308.11200

**Questions:**

See Weaknesses.

**Limitations:**

The authors have already described the limitation of this work in the paper, namely the lack of modeling multivariate relationships. To some extent, this is not a significant issue because many current cutting-edge studies have demonstrated that focusing solely on univariate temporal relationships can also be effective in multivariate tasks.

---

> ### Author Rebuttal · Authors · 2024-08-07
>
> We sincerely appreciate the comprehensive review, detailed feedback, and valuable suggestions from the reviewer Bhsw.
>
> > Q1: The major issue with this paper is the lack of analysis and comparison with significant literature. Please compare your method with SegRNN and clarify your differences and advantages.
>
> | **Advantages** | Information propagation paths | Whether a decoder is unnecessary | Directly capturing periodic semantic information |
> |:--------------:|:-----------------------------:|:--------------------------------:|:------------------------------------------------:|
> | **TPGN (PGN)** |              $O(1)$             |                 ✓                |                         ✓                        |
> |   **SegRNN**   |             $O(L/W)$            |                 ✗                |                         ✗                        |
>
> (1) Regarding information propagation paths: Shorter information propagation paths lead to less information loss, better captured dependencies, and lower training difficulty. In this regard, PGN can achieve an information propagation path of $O(1)$, while SegRNN, although capable of segment-wise iteration, still maintains an information propagation path of $O(L/W)$, where $W$ represents the number of segments. This is where PGN holds a greater advantage.
>
> (2) In terms of time series forecasting design: Although SegRNN adopts parallel prediction in the decoder stage, successfully addressing the error accumulation issue of iterative RNN predictions, our approach, TPGN, draws inspiration from Informer's design, omitting the need for a decoder module. This not only resolves the error accumulation problem but also results in a more concise model design.
>
> (3) Direct extraction of periodic semantic information: TPGN can directly extract semantic information related to the periodicity of time series data, while SegRNN segments the time series first and then feeds the representation of each segment into GRU units, which hinders direct capture of the periodicity within the time series.
>
> The segmented iteration and parallel prediction of SegRNN are indeed thought-provoking. We have conducted preliminary experiments and observed that TPGN (PGN) has shown some improvement compared to SegRNN. In the revised version, we will incorporate the above analysis.
>
> > Q2: In Section 2, you should distinguish between Linear-based and MLP-based methods. Methods like DLinear and FITS should be classified as Linear-based methods.
>
> Thanks again for your valuable suggestion. We will conduct a full check and modification in the revised version.
>
> > Q3: The description of HIE is unclear: (i) Figure 2(a) conflicts with Equation 1 in the paper, where H = HIE(Padding(X)), and the actual code. It is recommended to modify Figure 2 to make this clearer. (ii) Line 169 describes that “HIE(·) is a linear layer,” but in practice, the behavior of HIE is more like CNN (or TCN) rather than a sorely linear mapping. Given (ii), calling the proposed method RNN-based is debatable since it is more likely TCN-based.
>
> (i) Thank you for pointing out the issue. **We have made modifications to Figure 2 (a), and the new figure is shown as Figure A in the PDF file (newly submitted)**.
>
> (ii) Thanks again for your detailed review. In fact, HIE is indeed a linear layer. However, during our implementation, we were inspired by the design of methods like Transformer and iTransformer for the FFN part, and we also used conv1D to replace Linear.
>
> (iii) The function of the HIE layer is to replace the recurrent structure of RNN. In terms of specific code implementation, whether using conv1D or Linear, it does not affect its function of linear mapping. Additionally, the linear operation of HIE is just a part of the structure in PGN. Following HIE, there is the Gated Mechanism, and together they form the entirety of PGN. Therefore, PGN can be considered as the new successor to RNN-based methods.
>
> > Q4: You should include an ablation analysis of the normalization layer, explaining its impact on TPGN achieving state-of-the-art results.
>
> Okay, since the normalization layer was not used in the ECL, ETTh2, and WTH datasets, we focused on conducting ablation experiments on the normalization layer for the Traffic and ETTh1 datasets. **Due to space limitations, the experimental results have been placed in Table A of the PDF file. For details, please refer to Sec. A of the Global Rebuttal.**
>
> We observed a noticeable decrease in performance when the normalization layer was removed from the two datasets. This drop can be attributed to the inherent characteristics of the datasets. In Table 6, we calculated the distribution (Mean and STD) of the training and validation set for each dataset. The results indicate significant differences in the distributions of the Traffic dataset and the ETTh1 dataset. These differences can introduce disturbances to the model, hence the $norm$ should be set to 1 in this case. More details can be found in Appendix H.
>
> > Q5: It does not include the hyperparameter settings required to reproduce the key results in the paper. Are the hyperparameters in the main results all defaults? For instance, is TPGN_period=24? If not, providing a complete script file that can be run directly is necessary.
>
> For the 'TPGN_period', we were inspired by WITRAN and aimed to partition time series based on natural period. Therefore, for each task in the five datasets, 'TPGN_period' was set to 24. The selection of $norm$ varied depending on the dataset, as detailed in Table 6. The value of $dm$ for each forecasting task was determined through the validation set, with specific details available in Appendix D. **Due to space limitations, we have included the hyperparameters of our model for each task in Table B of the PDF file for result reproducibility, please refer to Sec. B of the Global Rebuttal for more details**. Furthermore, we will also enhance the files in the source code moving forward.

---

> > ### Comment · Reviewer_Bhsw · 2024-08-08
> >
> > Thank you for the detailed rebuttal; it addressed most of my concerns. The updated version should make the neccessary modifications mentioned in the rebuttal and include the analysis and results compared with SegRNN. I have increased my initial score.

---

> > > ### Author Response · Authors · 2024-08-08
> > > **Thanks**
> > >
> > > Thank you once more for your time and valuable suggestions. In the updated revisions, we will include the necessary modifications, analysis as mentioned earlier, and the results compared with SegRNN.

---

### Official Review · Reviewer_t9pC · 2024-07-10

**Soundness:** 3
**Presentation:** 3
**Contribution:** 3
**Rating:** 7
**Confidence:** 4

**Summary:**

This paper focuses on long-range time series forecasting problems. To address the limitations of RNNs, a novel paradigm called PGN is introduced as an alternative, providing shorter information propagation paths. Building upon PGN, the paper further presents a generic temporal modeling framework named as TPGN, which effectively captures both long-term and short-term periodic patterns, as well as local and global information, through a dual-branch design. The experimental results in this paper demonstrate that TPGN exhibits excellent performance in time series forecasting.

**Strengths:**

S1: This paper proposed a novel paradigm called PGN, which effectively tackles the inherent issues of RNNs through a simple yet powerful design. PGN exhibits a high level of innovation and holds the potential to replace traditional RNNs.

S2: TPGN primarily focuses on modeling the temporal dimension. Its dual-branch design makes sense as it captures both long-term and short-term periodicity, as well as the local and global characteristics of time series. Additionally, it is reasonable to set up different univariate forecasting tasks to evaluate TPGN's performance.

S3: This paper is well-written, and the presentation of the figures and tables is clear, making it easy to understand and follow. The experimental comparisons are comprehensive, including numerous advanced baseline models such as iTransformer, ModernTCN, FITS, TimeMixer, PDF, WITRAN, and Basisformer.

**Weaknesses:**

W1. For tables with a large amount of content, such as Table 1, it may be beneficial to consider using different colors for highlighting, as it could enhance clarity. Additionally, another option to consider is moving some of the experimental results to an appendix.

W2. While TPGN exhibits some advantages in terms of efficiency, I have noticed that it still appears to be challenging to reach the optimal level. Specifically, I have noticed that as the input sequence size increases, the efficiency of TPGN may gradually become inferior to that of iTransformer.

**Questions:**

Q1. Why was the Gated Mechanism designed in PGN this way in Figure 2 (a)? Can this part be replaced with GRU or other RNN variants?

Q2. Is it necessary to have two Linear layers in the design of the short-term branch in TPGN?

**Limitations:**

yes

---

> ### Author Rebuttal · Authors · 2024-08-07
>
> We extend our sincere appreciation to Reviewer t9pC for providing valuable feedback and acknowledgment of our research.
>
> > Q1: For tables with a large amount of content, such as Table 1, it may be beneficial to consider using different colors for highlighting, as it could enhance clarity. Additionally, another option to consider is moving some of the experimental results to an appendix.
>
> Thank you for your suggestion. In the revised version, we will enhance the table presentation by utilizing color changes, bolding, and other methods. Additionally, we will thoroughly check and optimize the entire paper.
>
> > Q2:  While TPGN exhibits some advantages in terms of efficiency, I have noticed that it still appears to be challenging to reach the optimal level. Specifically, I have noticed that as the input sequence size increases, the efficiency of TPGN may gradually become inferior to that of iTransformer.
>
> We have noticed that in terms of efficiency, linear-based methods such as FITS do indeed perform better. And for iTransformer, since it maps the entire sequence into a patch for processing the time dimension, while TPGN requires long-term and short-term information extraction branches in the time dimension, the efficiency of TPGN may slightly lag behind iTransformer when the input sequence length varies. However, TPGN still holds significant advantages as it can maintain good efficiency while ensuring optimal performance.
>
> > Q3: Why was the Gated Mechanism designed in PGN this way in Figure 2 (a)? Can this part be replaced with GRU or other RNN variants?
>
> In our design, only one gate is needed for information selection and fusion, which incurs smaller overhead compared to GRU's two gates and LSTM's three gates. While PGN, as a general architecture, can have its gated mechanism replaced with GRU and other RNN variants, this substitution would introduce higher costs. In terms of performance, we have observed that despite having only one gate, PGN still outperforms GRU or LSTM, more details of the experiments can be found in Table 2.
>
> > Q4: Is it necessary to have two Linear layers in the design of the short-term branch in TPGN?
>
> Yes, because in the short-term branch, the semantic information extracted by the two linear layers carries different meanings. The first layer primarily aggregates short-term information, extracting localized information. The second layer further aggregates the representation from the first layer at a higher level, extracting global information. While it is possible to extract global semantic information using a single linear layer, it would overlook the extraction of information at different granularity levels. Additionally, in terms of efficiency, the theoretical complexity of using only a single linear layer is $O(L)$, which is larger than the current $O(\sqrt{L})$.

---

> > ### Comment · Reviewer_t9pC · 2024-08-13
> >
> > Most of my concerns have been addressed, I will keep my positive recommendation.

---

> > > ### Author Response · Authors · 2024-08-13
> > > **Thanks**
> > >
> > > Thanks for your time and positive recommendation!

---

### Official Review · Reviewer_fqFX · 2024-07-12

**Soundness:** 2
**Presentation:** 4
**Contribution:** 2
**Rating:** 5
**Confidence:** 4

**Summary:**

The paper introduces a new model paradigm which aims to solve the traditional bottlenecks of RNN models, such as non-parallel computation, gradient explosion/vanishing issues, etc.

**Strengths:**

1. An important problem is studied in this paper.
2. The overall representation is clear and easy to follow.
3. A comprehensive summary of the related work is provided.

**Weaknesses:**

1. The overall contribution is not very significant.
2. Some questions regarding the time complexity and experiments need to be clarified.

**Questions:**

The model proposed in this paper is pretty neat and easy to follow. My questions mainly focus on the time complexity and experiments:
1. I think the total amount of computation done in the PGN should be O(L^2) because it is 1 + 2 + … + L, which is O(L^2). Although the PGN removes half of the computations of the self-attention, the self-attention and PGN still share the same asymptotic complexity. Thus, technically, replacing the PGN with self-attention won’t change the time complexity asymptotically. But I agree that it should be faster than RNN since it enables parallel computation as self-attention does. Also, this is the reason why I think the overall contribution is less exciting than the paper title claims. Can the authors kindly address this issue?
2. The above discussion can also be tested in the Efficiency of Execution experiment.
3. Regarding the experiments, only an input length of 168 is tested. Why did the authors choose to fixate on this input length instead of testing some other options?
4. In the ablation test, it seems that on some datasets (e.g., ETTh1), TPGN’s improvement is very slight compared to its LSTM/GRU/MLP ablations. Can the authors provide some analysis on these cases?
5. I am also interested to see some analysis on what would happen if the PGN is replaced by self-attention.

---

> ### Author Rebuttal · Authors · 2024-08-07
>
> We express our sincere gratitude to Reviewer fqFX for providing comprehensive review, insightful perspectives, and thought-provoking questions.
>
> > Q1: I think the total amount of computation done in the PGN should be O(L^2) ... Can the authors kindly address this issue?
>
> Sincerest gratitude for your detailed comments. Here we would like to provide more explanations about our model:
>
> Firstly, regarding the complexity analysis:
>
> (1) The computational complexity related to sequences in PGN mainly focuses on the HIE layer. In our specific implementation, we were inspired by the implementation details of methods like Transformer/iTransformer for the FFN part and used conv1D to replace Linear. Indeed, our actual convolution kernel (lookback window) size is set to $L-1$. However, typically in CNN computations, the convolution kernel size is considered a hyperparameter. For example, the convolution kernel in MICN[1] is actually linearly related to $L$, but the theoretical complexity of MICN remains $O(L)$. Therefore, when calculating the theoretical complexity in our case, we followed a similar approach.
>
> (2) The HIE layer can still be implemented using Linear. In this scenario, due to parameter sharing, the way each time step processes information from its previous $L-1$ time steps is exactly the same. Hence, we can follow the approach of WITRAN[2] in handling slices by expanding the dimensions that need to be processed in parallel and merging them into the sample dimension. In this case, Linear reduces $L-1$ dimensions to 1, so its theoretical complexity remains $O(L)$.
>
> (3) In our time series forecasting task, the lookback window size of PGN is set to $L-1$ to fully capture long-term information from the input signal. However, since PGN is a general paradigm, when applied to other tasks, the lookback window can be fixed to a certain value or adjusted. In such cases, regardless of whether implementation (1) or (2) is used, the theoretical complexity remains $O(L)$ or lower.
>
> (4) For self-attention, its computation between sequences is entirely different from PGN because it requires matrix operations to obtain information between every two time steps. Therefore, the necessary $O(L^2)$ theoretical complexity cannot be reduced using the aforementioned ways. From this, we can see that the essence of PGN is fundamentally different from self-attention.
>
> Secondly, In terms of innovation, we have focused on proposing PGN as the successor to RNN to address the issues stemming from RNN's excessive reliance on recurrent information propagation and overly long information pathways, including: (a) difficulty in capturing long-term dependencies in signals; (b) gradient vanishing/exploding; (c) low efficiency in serial computation.
>
> (1) Analysis of information propagation paths: PGN, through the HIE layer, can capture historical information at each time step with an $O(1)$ information propagation path, followed by information selection and fusion through Gated Mechanism, thereby altering RNN's $O(L)$ information propagation path. A shorter information propagation path lead to less information loss, better captured dependencies, and lower training difficulty. Consequently, PGN can more effectively capture long-term dependencies of inputs signals. And, as the information path shortens, it naturally resolves the issue of gradient vanishing/exploding.
>
> (2) Regarding efficiency, the thoughtful design of PGN allows for shared parameters in HIE, enabling information from subsequent time steps to be computed in parallel without waiting for the completion of computations from preceding time steps. This approach achieves good efficiency and resolves the inefficiency in serial computation of RNN.
>
> In conclusion, through its innovative approach, PGN addresses all the limitations of RNN. Therefore, we claimed that PGN can be considered as the new successor to RNN.
>
> **Refs [1][2] details are in Sec. G of Global Rebuttal.**
>
> > Q2: The above discussion can also be tested in the Efficiency of Execution experiment.
>
> Okay, thank you for your suggestion. We replaced PGN with self-attention and conducted efficiency experiments. **Due to space constraints, we have placed the results in newly submitted PDF file. More details please refer to Sec. C of the Global Rebuttal**.
>
> > Q3: Why did the authors choose to fixate on input length of 168 instead of testing some other options?
>
> This is because, on the one hand, considering factors such as the load capacity of training devices and data collection, forecasting longer futures with shorter sequence lengths is an important research problem, which has been the focus of much previous works, such as Informer, MICN, etc. On the other hand, in order for the model to have sufficient historical information for prediction from a limited sequence length, we aim for sequence length could be longer. Therefore, we chose 168 historical time steps (7 days) to predict the future (7 days, 14 days, 30 days, 60 days) as the forecasting tasks. This setting satisfies both considerations and serves as the basis for evaluating the performance of all models. Additionally, we also conducted experiments to demonstrate the reasonableness of our second consideration. **Due to space constraints, more details can be found in Sec. D in the Global Rebuttal**.
>
> > Q4: In the ablation test, some datasets ... Can the authors provide some analysis on these cases?
>
> Thank you again for your detailed question. We have conducted relevant statistics and analysis, but **due to space constraints, we regret to inform you that we present all the details in Sec. E of the Global Rebuttal**.
>
> > Q5: I am also interested to ... what would happen if the PGN is replaced by self-attention.
>
> Within the framework of TPGN, we replaced PGN with self-attention and conducted experiments on five datasets. **Due to space limitations, for experimental results and more detailed analysis, please refer to Sec. F in the Global Rebuttal**.

---

> > ### Comment · Reviewer_fqFX · 2024-08-12
> >
> > Thanks for the authors' response. My major concern regarding the time complexity has been addressed. I'll raise my score.

---

> > > ### Author Response · Authors · 2024-08-13
> > > **Thanks**
> > >
> > > Thank you once again for your time and valuable suggestions. We will include the above discussions in the revised version.

---

> > > ### Author Response · Authors · 2024-08-13
> > > **Thanks once more**
> > >
> > > After we sent comments earlier, we discovered a bug in the system's notifications as we did not receive any messages. However, we have noticed that this issue appears to have been resolved. Therefore, we sincerely thank you once more. Thank you once again for your time and valuable suggestions. We will include the above discussions in the revised version.

---

### Official Review · Reviewer_K1qz · 2024-07-14

**Soundness:** 3
**Presentation:** 3
**Contribution:** 3
**Rating:** 7
**Confidence:** 5

**Summary:**

This paper proposes a new network called PGN to capture the long-term dependencies of time series. Based on PGN, this paper further design TPGN for long-range time series forecasting. TPGN consists of two branches to respectively capture the long-term periodic patterns and short-term information of time series. Extensive experiments are conducted to show the effectiveness and efficiency of the TPGN.

**Strengths:**

S1. This paper is easy to follow. The motivations are clearly described by figures. The authors thoroughly analyze the information propagation modes of different time series forecasting models and explore new information propagation path to improve TS forecasting effectiveness.

S2. The design of the PGN is novel, which is a completely new network architecture and can effectively solve the inherent problems of classic RNN models. Both experimental results and theoretical analysis show the effectiveness and efficiency of PGN.

S3. This paper proposes TPGN upon PGN, which capture both the long-term and short-term characteristics of the time series with low computational complexity.

S4. Experiments are sufficient. Five benchmark datasets are evaluated and the most representative models proposed recently are included in the experiments.

**Weaknesses:**

W1. The computational complexity of TPGN is not well discussed in this paper, and it would be better if the inference efficiency was adequately discussed as the time series size increases.

W2. Some presentation needs to be improved. For example, it is difficult for readers to quickly get important conclusions on Table 1 and Table 4.

**Questions:**

In table of experiment comparison, could you explain why TPGN-long outperform TPGN in some cases.

**Limitations:**

Yes, the authors have fully discussed the limitations of their work.

---

> ### Author Rebuttal · Authors · 2024-08-07
>
> We sincerely appreciate Reviewer K1qz for offering valuable insights and recognizing of our work.
>
> > Q1: The computational complexity of TPGN is not well discussed in this paper, and it would be better if the inference efficiency was adequately discussed as the time series size increases.
>
> The TPGN architecture is designed to capture information more effectively from historical sequences. The complexity of processing input time series can be referenced in Sec. 3.3. For the forecasting module, our approach involves concatenating the local long-term and globally
> aggregated short-term information obtained by TPGN to predict the future at different time points within a period. As a result, the dimensionality of the forecasting module changes from $P \times (2 \times dm)$ to $P \times R_\mathrm{f}$, with a complexity of $O(R_\mathrm{f})$. As $R_\mathrm{f} \times P = L_f (L)$, the complexity of the forecasting module is $O(\sqrt{L})$. Indeed, it is also important to note that since both the input and output serires must be fully stored in memory, the total memory complexity should be $O(L)$. However, this does not conflict with the $O(\sqrt{L})$ complexity of TPGN. As the input seriers grows linearly, the complexity of TPGN also increases in a square root manner, with the actual time and memory overhead reflected in Figures 4(c) and (d). When the forecasting series grows linearly, although theoretically the complexity of TPGN should also increase in a square root manner, when $d_\mathrm{m}$ is relatively large, the overhead of the forecasting module is much smaller compared to the encoder part. Therefore, as shown in Figures 4(a) and (b), the actual cost variation is not significant as the forecasting sequence grows.
>
> > Q2: Some presentation needs to be improved. For example, it is difficult for readers to quickly get important conclusions on Table 1 and Table 4.
>
> Thank you for the reminder. In the revised version, we will carefully check the entire paper and enhance its presentation by adjusting colors, bolding text, particularly to emphasize the key conclusions in Table 1 and Table 4.
>
> > Q3: In table of experiment comparison, could you explain why TPGN-long outperform TPGN in some cases.
>
> From Table 2, we observe that in most datasets across various forecasting tasks, TPGN outperforms TPGN-long, with the exception of a few instances where TPGN-long performs better than TPGN in the Traffic dataset. Observing Figure 12, the Traffic dataset primarily exhibits strong periodicity along with short-term variations in different patterns. In this context, the long-term module in TPGN is already adept at capturing the predominant strong periodicity in sequences. However, in certain cases where the short-term variation patterns exhibit strong randomness, it may introduce disturbances to the short-term module of TPGN. Hence, there are scenarios in specific forecasting tasks where TPGN-long outperforms TPGN in terms of the MAE metric. Nevertheless, we also observe that TPGN remains best performance in terms of the MSE metric in these forecasting tasks.

---

> > ### Comment · Reviewer_K1qz · 2024-08-10
> > **Thanks for rebuttal**
> >
> > I have carefully read the rebuttal and all my concerns have been well addressed. I will accordingly raise my rate. Thanks.

---

> > > ### Author Response · Authors · 2024-08-10
> > > **Thanks**
> > >
> > > Thank you for your time and positive comments!

---

### Author Rebuttal · Authors · 2024-08-07

We sincerely thank reviewers for their thorough review and valuable suggestions.

# A  Ablation analysis of the normalization layer on the Traffic and ETTh1 datasets

**The experimental results can be found in Table A in the PDF file (newly submitted).** We observed a noticeable decrease in performance when the normalization layer was removed from the two datasets. This drop can be attributed to the inherent characteristics of the datasets. In Table 6, we calculated the distribution (Mean and STD) of the training and validation set for each dataset. The results indicate significant differences in the distributions of the Traffic dataset and the ETTh1 dataset. These differences can introduce disturbances to the model, hence the $norm$ should be set to 1 in this case. More details can be found in Appendix H.

# B Specific hyperparameters for reproducing our model

**We have presented the specific hyperparameters of our model for different tasks across all datasets in Table B of the PDF file (newly submitted)**, facilitating the direct reproducibility of our experimental results.

# C Efficiency of Execution experiment of replacing PGN with self-attention in TPGN

Within the TPGN framework, we replaced PGN with self-attention (referred to as TPGN-Attn). The results indicate that the overall practical overhead of TPGN-Attn is inferior to TPGN, **as shown in Figure B of the newly submitted PDF file.**

Specifically, following the setup outlined in Sec. 4.3, we conducted two sets of efficiency experiments on TPGN-Attn. In one set of experiments, we kept the input length fixed at 168 and varied the output length to 168/336/720/1440, while in the other set, we fixed the output length at 1440 and varied the input length to 168/336/720/1440. We integrated the results of TPGN-Attn with Figure 4, resulting in a new Figure B (For specific details, please refer to the newly submitted PDF file).

# D The experimental results for forecasting 1440 steps using 96 steps as input

We compared the performance of forecasting 1440 steps using 96 steps input to using 168 steps input. The results show that, in the 96-1440 scenario compared to the 168-1440 scenario, all models exhibited varying degrees of performance decline, with TPGN still demonstrating its advantage. This suggests that increasing the input length appropriately when predicting longer futures with shorter inputs provides the model with more information for improved performance. **The detailed experimental results can be found in Table C in PDF file (newly submitted).**

Specifically, we selected the three models ranked second in Table 1, WITRAN, iTransformer, and TimeMixer, and compared their performance in experiments predicting 1440 steps using 96 steps.


# E An analysis of the experimental results regarding TPGN and its LSTM/GRU/MLP ablations

We calculated the average improvements of TPGN compared to TPGN-LSTM/TPGN-GRU/TPGN-MLP across various tasks on each dataset. We found that the improvements were not as pronounced on the ETTh1 and Traffic datasets compared to the other datasets. Additionally, we also calculated the improvements of TPGN over the second-best model performance in Table 1 in terms of MSE across various tasks. We have summarized the results as follows. The table presents the average improvement of TPGN over the compared models across various tasks.

| Dataset | The second-best model | TPGN-LSTM | TPGN-GRU | TPGN-MLP |
|:-------:|:---------------------:|:---------:|:--------:|:--------:|
| Traffic |         9.38%         |   3.54%   |   3.33%  |  19.54%  |
|  ETTh1  |         3.79%         |   5.14%   |   5.52%  |   4.02%  |

(1) On the ETTh1 dataset, although the numerical improvement of TPGN over its LSTM/GRU/MLP ablations is not substantial, comparing TPGN with the second-best model reveals a significant degree of improvement. This discrepancy can be attributed to the characteristics of the ETTh1 dataset, making the improvement appear less pronounced compared to other datasets.

(2) On the Traffic dataset, TPGN shows less improvement over its LSTM/GRU ablations. This is due to the strong periodicity of the Traffic dataset, as observed in Figure 12. In this scenario, the TPGN framework plays a crucial role in capturing this strong periodicity effectively. Despite PGN in TPGN using only one gate, its performance surpasses that of GRU with two gates and LSTM with three gates, demonstrating that PGN can serve as the new successor to RNN.

(3) Furthermore, the results in the above Table reveal that the performance of the ablated variants of LSTM/GRU/MLP ablations across different datasets, indicating their limited adaptability to diverse domains. In contrast, TPGN demonstrates a greater advantage in this regard.

# F Forecasting performance of replacing PGN with self-attention in TPGN

Within the framework of TPGN, we replaced PGN with self-attention (called TPGN-Attn). The results show that TPGN exhibits a significant advantage over TPGN-Attn. **The specific experimental results are presented in Table D in the PDF file (newly submitted).**

Based on the results in the Table D, we calculated that TPGN outperforms TPGN-Attn in terms of MSE, with an average improvement of 26.97%. In particular, TPGN demonstrates an average reduction in MSE of 27.18% for the ECL dataset, 24.87% for the Traffic dataset, 6.99% for the ETTh1 dataset, 29.62% for the ETTh2 dataset, and 46.19% for the Weather dataset. The analysis above demonstrates the advantages of TPGN.

# G References

MICN[1]: Wang, H., et al. (2023). Micn: Multi-scale local and global context modeling for long-term series forecasting. In The eleventh international conference on learning representations.

WITRAN[2]: Jia, Y., et al. (2023). WITRAN: water-wave information transmission and recurrent acceleration network for long-range time series forecasting. In Proceedings of the 37th International Conference on Neural Information Processing Systems.

---

### Decision · Program_Chairs · 2024-09-25

**Decision:**

Accept (poster)

**Comment:**

The submission addresses the problem of lengthy information paths in RNNs by proposing a fully parallel gated model with O(1) path length. The approach is novel and well supported by discussions and empirical results. The rebuttal discussion led to a convergence in the reviewers' opinion, which broadly support acceptance of the work.